# Light-Dermo: A Lightweight Pretrained Convolution Neural Network for the Diagnosis of Multiclass Skin Lesions

**DOI:** 10.3390/diagnostics13030385

**Published:** 2023-01-19

**Authors:** Abdul Rauf Baig, Qaisar Abbas, Riyad Almakki, Mostafa E. A. Ibrahim, Lulwah AlSuwaidan, Alaa E. S. Ahmed

**Affiliations:** College of Computer and Information Sciences, Imam Mohammad Ibn Saud Islamic University (IMSIU), Riyadh 11432, Saudi Arabia

**Keywords:** pigmented skin lesions, deep learning, convolutional neural network, transfer learning, pretrained models, ShuffleNet, depthwise separable CNN

## Abstract

Skin cancer develops due to the unusual growth of skin cells. Early detection is critical for the recognition of multiclass pigmented skin lesions (PSLs). At an early stage, the manual work by ophthalmologists takes time to recognize the PSLs. Therefore, several “computer-aided diagnosis (CAD)” systems are developed by using image processing, machine learning (ML), and deep learning (DL) techniques. Deep-CNN models outperformed traditional ML approaches in extracting complex features from PSLs. In this study, a special transfer learning (TL)-based CNN model is suggested for the diagnosis of seven classes of PSLs. A novel approach (Light-Dermo) is developed that is based on a lightweight CNN model and applies the channelwise attention (CA) mechanism with a focus on computational efficiency. The ShuffleNet architecture is chosen as the backbone, and squeeze-and-excitation (SE) blocks are incorporated as the technique to enhance the original ShuffleNet architecture. Initially, an accessible dataset with 14,000 images of PSLs from seven classes is used to validate the Light-Dermo model. To increase the size of the dataset and control its imbalance, we have applied data augmentation techniques to seven classes of PSLs. By applying this technique, we collected 28,000 images from the HAM10000, ISIS-2019, and ISIC-2020 datasets. The outcomes of the experiments show that the suggested approach outperforms compared techniques in many cases. The most accurately trained model has an accuracy of 99.14%, a specificity of 98.20%, a sensitivity of 97.45%, and an F1-score of 98.1%, with fewer parameters compared to state-of-the-art DL models. The experimental results show that Light-Dermo assists the dermatologist in the better diagnosis of PSLs. The Light-Dermo code is available to the public on GitHub so that researchers can use it and improve it.

## 1. Introduction

Skin cancer is affecting the population and posing a significant financial burden on the global healthcare system. This is despite the fact that preliminary treatment may dramatically increase the cure rate for skin cancer. It is challenging due to the lack of access to dermatologists and the lack of training among other healthcare professionals [1]. The World Health Organization (WHO) anticipates that one individual in every three may suffer from skin cancer. The prevalence of skin cancer has been rising over the past few decades in nations including the USA, Canada, and Australia [2]. Early detection of skin cancer decreases the mortality rate. To identify early skin cancer, dermatologists use computer-aided diagnosis (CAD) systems, which were developed through machine learning (ML) and deep learning (DL) techniques. Recently, there is growing evidence that ML and DL can help dermatologists to make better clinical decisions. Studies have shown that DL algorithms can help doctors figure out the type of skin cancer [3].

Deep learning (DL) is a subset of machine learning that is particularly effective for image categorization tasks. DL uses multi-layer neural networks with numerous hidden layers made up of interconnected artificial neurons that process input data mathematically. A form of neural network called convolutional neural networks (CNNs) is particularly good at categorizing images. In practice, these are frequently used for image tasks. Since the CNNs replicate natural visual processing in the brain, which is capable of interpreting rich information such as the link between nearby pixels and objects, image categorization has been performed remarkably well [4] by CNN models. DL has recently made it possible to use diverse types of medical imaging to find abnormalities such as breast cancer, brain tumors, lung cancer, and skin lesions.

Several methods based on the pretrained architecture with a transfer learning (TL) scheme were also presented in the past to automatically classify two or seven classes of pigmented skin lesions (PSLs) when diagnosed through dermoscopy. This follows a general trend of building deeper and more complex networks to achieve higher accuracy. Some authors utilized modified CNNs or pretrained CNN models for the classification of PSL. Currently, embedded, IoT, and mobile devices are widely utilized. However, they frequently have low CPU and storage capacities. These devices cannot use more sophisticated networks because of the number of parameters and computational requirements. To meet the application requirements, it is possible to investigate unique network designs that provide the highest accuracy with an extremely constrained computational cost. To address these problems, compact and effective neural networks were developed, including SqueezeNet, MobileNet, and ShuffleNet [5]. These methods provide a compact architectural unit that may be integrated into existing networks to improve performance at a low computational cost.

In this study, we suggest innovative designs that pair the channelwise attention (CA) mechanism with effectively pretrained CNN networks to adhere to resource constraints (latency, memory size, etc.). We have selected the ShuffleNet [5] architecture as the backbones because of its effective topologies, which not only provide tiny networks but also allows for the encoding of more data. Squeeze-and-excitation (SE) blocks, which are regarded as CA mechanisms, are combined with the backbones to further increase the precision of pigmented skin lesions (PSLs) categorization. Comparative trials for the categorization of seven categories of PSLs demonstrate that this proposed lightweight CNN (Light-Dermo) model is more successful than its alternative structures. This outcome demonstrates that accuracy and efficiency have increased. Finally, the Light-Dermo can handle real-time applications.

### 1.1. Motivations

There are several reasons for developing a lightweight PSL multiple-class recognition classifier. Firstly, state-of-the-art approaches were mostly evaluated on a single dataset and classified two classes of PSLs (benign vs. malignant), compared to seven classes, but were computationally expensive. Overfitting and underfitting problems were mostly pointed out in past studies. There was a class imbalance problem in terms of multiple PSLs. To address these issues, this paper proposes a new lightweight pretrained model based on the ShuffleNet architecture which incorporates different layers, a separate nonlinear activation function such as GELU, and a channel shuffling technique.

### 1.2. Major Contributions

Below are the major advantages of Light-Dermo over alternative techniques to recognize seven classes of PSLs.

A new lightweight pretrained model based on the ShuffleNet architecture is developed in this paper which incorporates different layers, a separate nonlinear activation function such as GELU, and a channel shuffling technique.The Squeeze-and-excitation (SE) blocks are integrated to ShuffleNet architecture for developing a lightweight Light-Dermo model to manage resource constraints (latency, memory size, etc.).The Light-Dermo model has ability to reduce overfitting. The ShuffleNet network’s connections create quick pathways from the bottom layers to the top ones. As a result, the loss function (GELU) gives each layer more direction. Therefore, the dense connection protects against the overfitting problem better, especially when learning from insignificant amounts of data.The Light-Dermo model is a computational inexpensive solution for the diagnosis of multiple classes of pigmented skin lesions (PSLs) compared to state-of-the-art approaches, when tested on mobile devices.

The remainder of this paper is structured as follows. In Section 2, we have presented a brief introduction about previous works. Whereas in Section 3, we describe the background about the cutting-edge DL architectures based on TL schemes. The dataset and procedures used in this investigation are presented in Section 4. The findings and analysis of studies on the categorization of diseased pigmented skin lesions (PSLs) are presented in Section 5. The key results are described in Section 6, and finally, this paper concludes in Section 7.

## 2. Literature Review

Several DL-based CAD systems were developed in the past to help ophthalmologists better diagnose multiclasses of PSLs. Before 2016, many studies were developed based on the standard steps such as preprocessing, segmentation, feature extraction, and classification. Different skin lesion datasets are now available to the public. To discriminate among PSLs, researchers have created DL algorithms. Over time, it was noticed that the CNNs had more effective features compared to hand-crafted methods. In this paper, we have studied and selected past studies (as shown in Table 1) that have recently utilized the DL and TL-based methods for the diagnosis of PSLs. Those studies were described in brief in the subsequent paragraphs.

A transfer learning (TL)-based approach is described in paper [5] for the classification of only melanocytic skin lesions. Using the publicly available ISIC-2021 dataset from Kaggle, the authors tested their approach. The suggested model had a 98.35% success rate. Another classification method was presented in paper [6]. Using a few convolutional neural networks (CNNs) trained on the HAM-2010 dataset, this method predicts seven different classes of PSL lesions. The authors used nine CNN models, which were experimentally selected. Then, a weight matrix was created by integrating the decisions of each CNN model into a decision fusion module. They examined the performance of each of CNN’s network to design a multi-network system. For each network, a 75% performance threshold has been achieved. Whereas in the study of [7], the authors recognized PSLs in two categories; benign and malignant. The proposed models were trained on the ISIC-2020 dataset. Using three pre-trained models, they obtained an average classification accuracy of 98.73%.

In study [8], the authors present a DL approach that combines a TL scheme with the pre-trained AlexNet. Based on the results, it was concluded that the suggested method is quite successful in categorizing skin lesions into seven classes. The achieved percentages for accuracy (ACC), sensitivity (SE), specificity (SP), and precision (PR) are 98.70%, 95.60%, 99.27%, and 95.06%, respectively. Another DL-based algorithm is proposed in paper [9]. By recognizing two categories of PSLs, the accuracy was equal to that of dermatologists. Authors in [10] provided methods for data augmentation to balance different types of lesions. The proposed model was found to be effective in classifying and identifying PSLs by using the LSTM and MobileNet V2 approaches.

In paper [11], an enhanced capsule network for dermoscopic picture classification (FixCaps) is introduced. FixCaps uses a large, high-performance kernel at the bottom convolution layer. The loss of spatial information brought on by convolution and pooling was lessened by using the convolutional block attention module. Based on the results of the experiments, FixCaps is better at finding skin cancer than IRv2-SA, which had an accuracy rate of 96.49% on the HAM10000 dataset. For the work in [12], the authors implemented a preprocessing image pipeline. To satisfy the needs of each model, they augmented the dataset to eliminate hairs from the images. The EfficientNet B0-B7 is trained on the HAM10000 dataset by presenting a transfer learning (TL) algorithm. An F1 score of 87% and a Top-1 accuracy of 87.91 percent were presented. In paper [13], the authors showed how quickly and effectively nonlinear activation functions work on a CNN when just a few picture datasets are available. According to experimental findings, the proposed model performs better than the model that divides skin lesions into three classes of PSLs. A lightweight and effective model is suggested for the precise categorization of skin lesions [14]. To get the best results, layers with dynamic-sized kernels are utilized, leaving relatively few trainable parameters. Furthermore, the suggested model intentionally uses both the leaky-ReLU and ReLU activation functions. The HAM10000 dataset’s classes were correctly categorized by the model. The model outperformed other state-of-the-art heavy models with an overall accuracy of 97.85%.

The authors in [15] presented a CNN model based on DL for accurately distinguishing benign from malignant skin lesions. Three steps make up the suggested model. Preprocessing, normalization, and augmentation. To evaluate the proposed DCNN model’s performance, AlexNet, ResNet, VGG-16, DenseNet, MobileNet, and other transfer learning techniques are compared with it. The model was examined using the HAM10000 dataset, and the findings revealed that it had the highest training accuracy of 93.16% and testing accuracy of 91.93%. The results of the proposed DCNN model show that it is more accurate and reliable than other methods. In paper [16], the authors developed a differentiation system to classify two classes of PSLs. The authors used one flat layer, two dense layers with LeakyReLU, and another dense layer with the sigmoid function. On average, they obtained 89% ACC. The work in [17] investigated the classification of clinical images of four skin disorders using a DL algorithm. Images from the training set of three datasets were used to adjust the ResNet-152 convolutional neural network model. The areas under the curve were 0.96, according to the results.

To improve classification accuracy, the authors in [18] merged the deep feature vector and the colored network feature vector using a parallel fusion technique. Machine learning classifiers correctly divide the dermoscopic pictures into two groups, benign and malignant melanoma, using this optimized fused feature vector as input. Three skin lesion datasets were used. The maximum classification accuracy was 99.8% for the ISBI-2016 dataset.

In paper [19], authors suggested a DL method for categorizing seven different forms of skin cancer. They made use of the HAM10000 datasets and used augmentation to increase the dataset size. They concluded that, when compared to other machine learning techniques used in the proposed work, convolutional neural networks offer superior accuracy. They achieved an accuracy of 95.18%. Using a CNN, authors in [20] offered an automated method for skin lesion detection and identification. The findings support the notion that the suggested method performs better than several other existing methods and achieved an accuracy of 98.4% on the PH2 dataset. The efficiency of skin lesion categorization using CNN-TL is impacted by image size, according to paper [21]. They also demonstrated that, in terms of performance, image cropping is superior to image resizing. Finally, the best classification performance is displayed by using a straightforward ensemble technique that combines the results from images cut at three fine-tuned CNNs and six different scales. Recent studies suggested that DL-based algorithms [22] presented a potential answer to recognizing images for darker skin tones and diversifying image repositories dominated by light skin. As a result, they created a DL-based technique for training and evaluating various photos that accurately depict human skin tones. However, skin cancer is more common in white-skinned people than in black-skinned people.

To solve the research issues, a multi-class multi-level (MCML) [23] classification algorithm inspired by the “divide and conquer” rule was investigated. The MCML classification algorithm was developed utilizing both conventional and sophisticated machine learning techniques. They presented improved strategies for noise reduction in the conventional machine learning methodology. The suggested technique uses a DL-based, pre-trained CNN model, which then uses the segmented color lesion pictures to extract features. To choose the most discriminant features and escape the dimensionality curse, they used the enhanced moth flame optimization (IMFO) approach. The resulting characteristics achieved an accuracy of 95.38%. The authors in paper [24] describe another method for extracting features from segmented color lesion images using a pre-trained CNN model. On the ISBI 2016, ISBI 2017, ISIC 2018, and PH2 datasets, the segmentation performance of the suggested technique is examined, with accuracy values of 95.38%. The HAM10000 dataset was used to examine the classification performance, which resulted in an accuracy of 90.67%.

In one of our studies [25], we developed a perceptually oriented color space to extract visual features. Additionally, a new DermoDeep system based on a five-layer architecture is suggested. Based on 2800 PSLs, comprising of 1400 nevi and 1400 malignant lesions, this DermoDeep system was achieved with a ACC of 0.96, an SE of 93%, and an SP of 95%. Whereas in paper [26], the binary classification is used by utilizing the multi-class confidence values supplied by the other half of the network. While any CNN architecture might be utilized for both challenges, they have selected the Inception-v3 pretrained model for both classification tasks. The entire network is trained in the conventional manner, and the multi-class accuracy is noticeably improved (by 7% when compared to the balanced multi-class accuracy). To increase effectiveness and performance, the authors in paper [27] suggested the Deep-CNN model, which was constructed with several layers, various filter sizes, and fewer filters and parameters. For experiments, dermoscopic pictures are obtained from the different versions of the ISIC dataset. It achieves 94% accuracy, 93% sensitivity, and 91% specificity in ISIC-17. Likewise [28], a unique parallel fusion method is utilized to combine OCFs with a DCNN-9 model to extract deep features. After that, the most reliable characteristics for classification are chosen using a normal distribution-based high-rank feature selection approach. Datasets from the ISBI series (2016, 2017, and 2018) are used to assess the recommended technique. On all three datasets, the provided technique achieved classification accuracy of 92.1%, 96.5%, and 85.1%, respectively, demonstrating its impressive performance.

**Table 1 diagnostics-13-00385-t001:** State-of-the-art studies to diagnosis PSLs by using pretrained TL models.

Ref.	Description	* Classification	Dataset	Augment?	Results%	Limitations
[8]	Pigmented Skin Lesions (PSLs) classify into seven classes by using TL approach.	AlexNet	ISIC2018	Yes	ACC.: 89.7	Evaluated on single dataset, classify seven-classes, but computational expensive, classifier overfitting.
[10]	The combine work of TL and DL.	MobileNet V2 and LSTM	HAM10000	Yes	ACC: 85	Classes imbalance problem, binary classification, tested on signal dataset and computational expensive.
[11]	The approach employs FixCaps for dermoscopic image classification.	FixCaps	HAM10000	Yes	Acc.: 96.49	Classes imbalance problem, evaluated on single dataset, classify only two-classes, and computational expensive.
[12]	The approach is a multiclass EfficientNet TL classifier to recognize different classes of PSLs.	EfficientNet	HAM10000	Yes	ACC.: 87.91	Single dataset, accuracy is not good, which limits the detection accuracy, 6 classes only, overfitting and computational expensive
[13]	They developed another technique based on CNN and non-linear activations functions.	Employ Linear and nonlinear activation functions either the hidden layers or output layers	PH2 ISIC	Yes	ACC: 97.5	Evaluated on two datasets, classify only two-classes, classifier overfitting, and Computational expensive.
[14]	CNN model along with activation functions.	Multiple-CNN models	HAM10000 Dataset	Yes	ACC: 97.85	Three classes of PSLs and reduced hyper-parameters so computational expensive, classifier overfitting and used only single dataset.
[15]	The approach is based on convolutional neural network model based on deep learning (DCNN to accurately classify the malignant skin lesions	DCNN	HAM10000 Dataset	Yes	ACC:91.3	Classes imbalance problem, evaluated on single dataset, classify seven-classes, and computational expensive, classifier overfitting.
[16]	Differentiate only benign and Malignant.	VGG16	Skin Cancer: Malignant vs. Benign ^1^	Yes	ACC: 89	Classes imbalance problem, binary classification only two-classes, not generalize solution, and computational expensive.
[17]	The approach of the classifier is based on Deep Learning Algorithm	ResNet-152	ASAN Edinburgh Hallym	Yes	ACC: 96	Classes imbalance problem, evaluated on single dataset, classify only two-classes, and computational expensive.
[18]	This technique uses features fusion approach to recognize PSLs.	PDFFEM	ISBI 2016, ISIC 2017, and PH2.	Yes	ISBI 2016 ACC:99.8	Image processing, handcrafted-based feature extraction approach, which limits the detection accuracy, 6 classes only, classifier underfitting, and computational expensive
[19]	Different classifiers are utilized to evaluate the approach.	DT, KNN, LR and LAD	HAM10000	Yes	ACC:95.18	Classes imbalance problem, evaluated on single dataset, classify seven-classes, and computational expensive, classifier overfitting.
[20]	The approach is a Heterogeneous of Deep CNN Features Fusion and Reduction	SVM, KNN and NN	PH2, ISBI 2016, ISBI 2017	Yes	ACC: 95.1%	Image processing, handcrafted-based feature extraction approach, which limits the detection accuracy, 6 classes only and computational expensive
[22]	Segmentation and Classification of Melanoma and Nevis	KNN, CNN	ISIC, DermNet NZ	Yes	--	Classes imbalance problem, evaluated on two datasets, classify only two-classes, and computational expensive.
[24]	Segmentation and classification approach	Pretrain CN, moth flame optimization (IMFO)	ISBI 2016, ISBI 2017, ISIC 2018, and PH2, HAM10000	Yes	ACC: 91%	Image processing, handcrafted-based feature extraction approach, which limits the detection accuracy, 6 classes only and computational expensive
[25]	Dermo-Deep is developed for classification based on two classes	five-layer pretrained CNN architecture	ISBI 2016, ISBI 2017, ISIC 2018, and PH2, HAM10000	No	ACC: 96%	Classes imbalance problem, classify seven-classes, and computational expensive, classifier overfitting.
[26]	Classification of seven classes to recognize PSLs	Google’s Inception-v3	HAM10000	Yes	ACC: 90%	Classes imbalance problem, binary classification only two-classes, and computational expensive.
[27]	A DCNN model is developed, which was constructed with several layers, various filter sizes, and fewer filters and parameters	DCNN	ISIC-17, ISIC-18, ISIC-19	Yes	ACC: 94%	Classes imbalance problem, classify only two-classes, and computational expensive.
[29]	Different pretrained models based on transfer learning techniques were evaluated in recent study	DenseNet201	ISIC	Yes	---	Limits the detection accuracy, 6 classes only and computational expensive

* DT: decision tree, KNN: k-nearest neighbor, PDFFEM: pigmented deep fused features extraction method, LDA: Linear Discriminant Analysis, SVM: support vector machine, NN: neural network, ResNet: residual network, DCNN: deep convolutional neural network, FixCaps: improved capsule network, LSTM: long short-term memory, ^1^ https://www.kaggle.com/datasets/fanconic/skin-cancer-malignant-vs-benign (accessed on 2 September 2022).

Different pretrained models based on transfer learning techniques were evaluated in a recent study [29]. The dataset utilized for these tests contains 10,154 images from ISIC 2018. The findings demonstrate that DenseNet201 outperforms other models with an accuracy of 0.825 and enhances skin lesion categorization under various disorders. The suggested study displays the distinct factors, including the precision of all pretrained learning networks, that contributed to the development of a successful automated classification model for a variety of skin lesions. In paper [30], the ML-basic models (logistic regression, SVM, random forest, KNN, and gradient boosting machine) were trained using manually created features. These base models’ predictions were utilized to train the level-one model stacking on the training set using cross-validation. Transfer learning was carried out using pre-trained deep learning models (MobileNet, Xception, ResNet50, ResNet50V2, and DenseNet121) using ImageNet data. The DL-based models were then evaluated after being ensembled with various model combinations. The experimental results indicate that the authors achieved good accuracy for the classification of PSLs.

## 3. Background of Cutting-Edge TL Networks

A broad range of deep neural network models, including AlexNet, VGG, GoogleNet, and ResNet, have been proposed because of improvements in hardware technology development [31,32,33,34,35,36,37]. Due to the high volume of parameters and computational processes, these deep and broad networks slow down training and detection. The most cutting-edge networks are taken into consideration in this study, including Inception, ResNet, Xception, AlexNet, SqueezeNet, MobileNet, and ShuffleNet [38]. To increase representational capacity and decrease computing complexity, they have different architectural designs.

### 3.1. AlexNet TL

Five convolutional layers and three fully linked layers make up AlexNet [32]. Following the last fully connected layer, the softmax layer generates the distribution of class labels. Usually, normalizing and pooling layers come after the convolution layers as an alternative. ReLU nonlinear activation units often mix with most of the layers in this design. Max pooling’s purpose is to reduce the size of the feature map (downsampling). A dropout layer is connected to the first two completely connected layers as well. In trials, the last completely linked layers in the pigmented skin lesions (PSLs) datasets were changed to two classes.

### 3.2. MobileNet TL

The Google research team suggests MobileNet [33] is a mobile-first computer vision model. The fundamental principle of MobileNet is to drastically reduce both the size of the model and the computational complexity by using depthwise separable convolution. The two types of convolutions that make up a depthwise separable convolution are pointwise and depthwise. Pointwise convolution is used to modify the dimension, whereas depthwise convolution applies a single filter to each input channel. Therefore, depthwise separable convolutions work well with good accuracy and speed and can reduce computation by eight to nine times. Finally, the model may be successfully used with portable electronics.

### 3.3. SqueezeNet TL

For computer vision applications, the SqueezeNet [34] architecture is a deep CNN that focuses on efficiency (having fewer parameters and a smaller model size). The fire module is the fundamental component of the SqueezeNet design. The module has both an expansion phase and a squeeze phase. A series of 11 filters are applied during the squeeze phase, and then a ReLU activation follows. There are always fewer learned squeeze filters than there is input volume. As a result, the squeeze phase may be thought of as a procedure that reduces the number of dimensions, while also capturing pixel correlations across input channels. The squeeze phase’s output is used to train a mixture of one and three convolutions in the expanding phase. The spatial correlations between pixels are captured using the bigger 3D filters. The outputs of the expanding phase are then checked by a ReLU activation, which adds them up over the channel dimension. The number of filters used in the expanding phase was suggested to be between one and three, as many as were used in the squeeze phase. The traditional convolution layers, max pooling, fire modules, and an average pooling layer at the end are stacked to create the whole SqueezeNet architecture. There are no fully linked layers in the model.

### 3.4. Inception TL

The Inception network [35] is a deep convolutional architecture that was first released in 2014 as GoogLeNet. The design has been improved in a variety of ways, such as by factoring convolutions with bigger spatial filters for computational efficiency and by using batch normalization layers to speed up training. We chose the Inceptionv3 model because of its exceptional capabilities for object localization and image identification. The Inception module, which comes in a variety of forms, is the essential building block for all Inception-style networks. After accepting input, the module branches into four separate directions, each of which does a distinct series of actions. The input goes via a pooling process, followed by convolutional layers with varying kernel sizes (1 × 1, 2 × 2, and 3 × 3, respectively). The module can recognize complicated patterns at various scales by using different kernel sizes. All branches’ outputs are concatenated channel-wise. Early on in the network’s development, the Inception-v3 network’s general architecture was made up of typical three-layer convolutional layers, some of which were followed by max pooling operations.

### 3.5. ResNet TL

Deep residual networks (ResNet) [36] are a series of exceptionally deep CNN architectures that are utilized for object identification, localization, and image recognition. The fact that the winning network had 152 layers validates the positive effects of network depth on visual representations. However, vanishing gradients and performance deterioration are two significant issues that arise while training networks with increasing depth. By using skip connections, the authors were able to solve the issues and prevent information loss as the network got deeper. The residual module of the two variations is the foundation for building deep residual networks. Two convolutional layers make up the left route of the residual module. These layers use three kernels and maintain the spatial dimensions. Additionally, the ResNet model used batch normalization (BN) and rectified linear units (ReLU) activation functions. The input is further added to the output of the left path on the right path, which is the skip connection. As a result, the ResNet model is successfully utilized in many models to skip connections.

### 3.6. Xception TL

Another deep CNN architecture is called the Xception [37] model. It makes use of the residual connections suggested by ResNet models and was inspired by the Inception architecture. However, depthwise separable convolution (Depth-CNN) layers are used in place of the Inception modules. A depth-first CNN consists of a pointwise convolution (1 × 1) to map the cross-channel correlations after a depth-first convolution (spatial convolutions of 3 × 3, 5 × 5, etc.) is performed over each input channel to map the spatial correlations. The Xception design only relies on convolution layers that can be separated by depthwise convolution, and it strongly assumes that cross-channel correlations and spatial correlations can be mapped separately. The network has 14 modules made up of 36 convolutional layers. Except for the first and last modules, all modules have residual connections. For a detailed explanation of the model specification.

### 3.7. ShuffleNet TL

The ShuffleNet model [38], which significantly lowers the computation cost, addresses the rising need for operating effective DL networks on embedded devices while maintaining accuracy. For the ImageNet classification challenge, their model outperformed MobileNet with a smaller top-1 error. To extract more feature maps in the standard version of the ShuffleNet TL model, the authors used pointwise convolution in conjunction with the channel-shuffle operation. These steps enable the network to encode more information, which is typical of the model. The goal of pointwise group convolution is to use dense (1 × 1) convolutions to reduce computation costs. Group convolutions deteriorate representation and block information because only a small percentage of the input channel is used to create the outputs from a specific channel. The channel-shuffle operation solution is offered as a means of resolving this problem. The purpose of the channel shuffle is to get input data from various groups by dividing the channels in each group into many groups, which are then sent into the subsequent layer as various subgroups. In this paper, we have improved the ShuffleNet as described in the methodology section.

## 4. Materials and Methods

This section is used to describe in detail the dataset used for the study and the proposed methodology.

### 4.1. Data Acquisition

The Harvard database made the 10,015 dermatoscopy images in the HAM10000 (“Human Against Machine with 10,000 Training Images”) dataset [39] available to the public with the aim of providing training data for automating the process of classifying skin cancer lesions. The goal of this research was to give the public access to a large and diverse data source for machine learning training so that the outcomes could be compared to those of human experts. This dataset includes the following 7 kinds of skin cancer lesions such as “actinic keratoses (AK), basal cell carcinoma (BCC), benign keratosis (BK), dermatofibroma (DF), melanoma (MEL), melanocytic nevi (NV), and vascular skin lesions (VASC)”. Figure 1 is a visually represented example of a patient’s history and type of pigmented skin lesions (PSLs) in the HAM10000 dataset. Figure 2 shows an example of selecting a region-of-interest (ROI) from PSL lesions (6000 images) after the area was automatically chosen from the centroid.

In addition, we utilized the ISIC-2019 [40] dataset, which consists of nine classes of PSLs. This dataset contains a total of 33,126 PSL images. We have selected only seven classes from the ISIC-2019 dataset, such as actinic keratoses (AK), basal cell carcinoma (BCC), benign keratosis (BK), dermatofibroma (DF), melanoma (MEL), melanocytic nevi (NV), and vascular skin lesions (VASC). The Kaggle platform’s competition includes a download link for this dataset. We have selected only 6000 PSLs from the ISIC-2020 images. Recent ISIC-2020 developments in the field have revealed a growing emphasis on the models capable of multi-class predictions [41,42]. From these datasets, there were 25,331 PSL images available based on nine classes. However, we have only selected seven classes, which consisted of 2000 PSLs. In total, we have used 14,000 PSLs from these datasets and removed many duplicate lesions. In Figure 3, the distribution of samples is presented from the ISIC-2019 and ISIC-2020 datasets according to the patients’ history and type of pigmented skin lesions (PSLs). In addition, Table 2 represents the total images in each class, which were selected from the HAM10000, ISIC-2019, and ISIC-2020 datasets.

### 4.2. Proposed Methodology

Figure 3 depicts a simplified attentional-convolutional network-based learning (Light-Dermo) framework for the recognition of PSLs. The Inception-v3 model is a transfer learning (TL) model that is included in four stages of ShuffleNet model, such as stage 1, stage 2, stage 3, and stage 4. These stages are listed in that order, starting from the bottom block to the top block. In this study, we used the same stages which were selected by empirical result analysis. We have utilized four stages, but in a different formation compared to original ShuffleNet model. However, compared to the CNN [29] model, we have used depthwise and pointwise separable CNNs (DPS-CNN). In practice, the DPS-CNNs require fewer parameters and adjustments than regular CNNs, which reduces overfitting. They are appropriate as well for mobile vision applications because they need fewer computations, which makes them more affordable computationally. In addition, we used the GELU activation function instead of other regular non-linear functions and different layers [29]. Each block is made up of (1 × 1), (2 × 2), and (3 × 3) convolutional layers with 64, 128, 256, and 512 filters, respectively. The stride is one for each of the four blocks. An SE block comes after each block. The BN layers and the ReLU come before each convolutional layer. A max pooling layer with a filter size of (2 × 2) and a stride of 2 is followed by SE blocks 1 through 4. Figure 4 has displayed the proposed architecture for an improved ShuffleNet deep learning model in terms of activation function and layers of the network. A global average pooling layer is followed by SE block 4, fully connected (FC), GELU, and then a softmax layer is used to avoid overfitting. The output layer, which has seven neurons encoding seven classes of pigmented skin lesions (PSLs), is the last layer.

#### 4.2.1. Data Imbalance and Augmentation

We discovered a considerable difference in the number of images contained inside the various classes of PSLs in the ISIC-2019, ISIC-2020, and HAM10000 datasets. In contrast to the other classes, the melanocytic nevi (NV) class, for instance, has a lot of samples. Additionally, the classifications of dermatofibroma (DF) and vascular lesions (VASC) contain fewer samples. To effectively train a DL-based model, we need enough balanced data, as illustrated in Figure 1 and Figure 3, which display the bar representation of the class-wise sample distribution of the original dataset. Also, Table 2 shows the overall 14,000 PSLs. To prevent biased training of the DL model, data balancing is carried out. Additionally, the imbalanced data may lead the model training to stay biased towards some classes with a relatively significant number of examples. Therefore, we applied augmentation techniques to balance our 14,000 PSLs images. We expanded our dataset by more than 14,000 and balanced it for each class using data augmentation, which consists of seven classes.

In this study, we applied image transform steps such as cropping, resizing, adding a random amount of Gaussian noise, adjusting brightness with contrast, rotating by 30°, and horizontal and vertical flipping to increase the training set size and reduce overfitting. All images were resized to a uniform (256 × 256 × 3) size. A visual example of the data augmentation technique is displayed in Figure 4, and parameter settings are mentioned in Table 3. By applying these transformations, we have obtained 28,000 PSLs consisting of seven classes, and each class has 4000 PSL images. A visual example of this transformation step is described in Figure 5.

#### 4.2.2. Improved ShuffleNet-Light Architecture

Overall steps of proposed ShuffleNet-Light architecture for Features Extraction and Classification of PSLs are displayed in Algorithm 1. Recent works in a variety of fields, such as natural language processing (NLP), image captioning, and image understanding, frequently use the attention mechanism. With minimal computational overhead, the SE introduces the SE block (see Figure 4), which self-recalibrates the feature map via channelwise significance to improve the performance of current state-of-the-art models. The SE block may be immediately applied to current network designs because of its flexibility. The SE block is integrated into ShuffleNet. The first is called the SE-Inside block, and it layers the SE unit straight after the final convolutional layer to embed it inside the ShuffleNet unit. The SE unit is positioned after the summation (“Add,” “Concat”) with the identity branch in the SE-Post block method to sum up. Due to the computational expense, if we use the tactics in all 16 blocks of ShuffleNet, we will need a lot of SE units. Without a doubt, the expensive density will force the network to become redundant.

In this study, the depth feature map produced by CNN is re-calibrated via the channel attention technique to enhance the model’s feature extraction performance and accomplish accurate classification of PSLs. Through SE operation, the network may choose to amplify useful feature channels and suppress unnecessary feature channels to reduce redundant information. Different weight ratios can be assigned from a global information perspective. By learning, the SE block automatically learns the weight of each feature channel and adjusts the weight of the original feature channel. In Figure 6a, the SE block structure is displayed. The suggested structure does not require a significant increase in computational cost because it accepts each feature map following the “concat” operation as an input. The feature refinement improves the model’s capacity for learning. Figure 6 depicts the entire classification structure of pigmented skin lesions (PSLs).

To implement the final approach, we only use SE blocks at the conclusion of the first four stages of the architecture (a total of four stages, each including multiple blocks), which results in the addition of just four SE blocks to each of the two structures. Figure 4 shows the network designs of enhanced ShuffleNets. The SE blocks are depicted as the lower portions of the frameworks in Figure 6a,b. In the squeeze phase, global average pooling is used to provide channel-wise statistics that embed global spatial information. The simplest aggregation method, global average pooling, is used to specifically reduce the spatial dimensions of the entire image.

The branch normalization (BN) layer’s benefits include accelerating training and enhancing the network’s generalization capacity. It can take the place of the normalized local response layer and function effectively as a normalized network layer. We concentrate on one activation for the sake of clarity because normalization operates differently for each activation. Additionally, the aggregated information produced from the squeeze information in the excitation phase is used in a bottleneck with two tiny, completely linked layers. A set of per-channel modulation weights is then produced when the excitation process completely captures channelwise interdependence. The input feature maps are given these weights. The gating mechanism is the subject of this operation, which is known as feature recalibration. To accomplish this, a straightforward gating technique with softmax is used to finally classify PSL lesions.
**Algorithm 1**: ShuffleNet-Light Architecture for Features Extraction and Classification of PSLs***Input:****Input Tensor (*X*), 2-D of (256 × 256 × 3) PSLs training dataset.****Output:****Obtained and Classified feature map*x=(x1,x2,…,xn)*augmented 2-D image****Main Process:****Step 1. Define number of stages = 4**Step 2. Iterate for Each Stage*(*a*)*“Depthwise-CNN is applied to tensor x by kernel size of (3 × 3), which includes a number of filters; branch normalization, the ReLU activation function, Pointwise-CNN by kernel size of (1 × 1), branch normalization, and the GELU” activation function are applied.*(*b*)*“Pointwise-CNN is applied to tensor x by kernel size of (1 × 1), which includes a number of filters, branch normalization, ReLU activation function, Pointwise-CNN by kernel size of (1 × 1), branch normalization, GELU” activation function are applied.**Step 3. Fscale = Squeeze and Excitation (SE) block contains expansion (1 × 1 × 3) layers.**Step 4. Fcat(i) = concatenation (# features-maps)**Step 5. channel = shuffle (x)**[End Step 2]**Step 6. Model Construction*(*a*)*Define Global-average-pooling layer*(*b*)*Define Fully-Connected (FC) Layer and applied GELU function.**Step 7. Afterward, the feature map*x=(x1,x2,…,xn)*generated, which is recognized by Softmax function.**Step 8. Test samples*yit *are predicted to the class label using the decision function of the below equation.*yit=∑t=0M−1ft(xi)

#### 4.2.3. Design of Network Structure

We process the channel information using three technologies, such as depthwise separable convolutional (DConv), channel-shuffle (CS), and squeeze and excitation (SE) modules, which improve the correlation between different channel information and can filter noisy channel information. The channel information improvement element significantly improves the classification model’s accuracy and stability.

The DConv step only uses one convolution kernel (1 *×* 1) per channel, and each channel is convolved only once at a time. In other words, it is the same as treating the input feature map like a single-channel image. Because of the multi-channel nature of pigmented skin lesions (PSLs), DConv is more appropriate than ordinary convolutions. The standard convolution is only applied to the input image through a convolution kernel in the DConv. The value obtained by multiplying the original data’s weight and the convolution kernel’s corresponding position is mapped to the corresponding position of the feature maps. In other words, all input channels’ data is processed by each convolution kernel. Each convolution kernel in a normal convolution processes the data from every input channel and turns it into a feature map. The traditional convolution (Conv)’s parameter amount, ConvP, is calculated using the following formula:(1)ConvP = ConvW × ConvH ×Conv × Conv 

Among them, the ConvW, ConvH, and Conv stand for the number of convolution kernels, their size, and the quantity of input data channels, respectively. The basic idea behind DConv is that each convolution kernel processes each input channel individually to produce a map, as illustrated in Figure 7. By processing just 2-dimensional spatial data in this way, the convolution kernel reduces the processing of data from many channels. As a result, the following is the calculation formula for the parameter amount DConvP of the DConv:DConvP = ConvW × ConvH × Conv(2)

When compared to the traditional convolution approach, DConv requires fewer parameters for the calculation and, to some extent, increases the model’s operational efficiency. But since DConv doesn’t care about information across channels, which would mean that useful information would be lost, the channel-shuffle technique is used to make up for it.

The channel-shuffle compensates for the DConv’s shortcomings. Each convolution handles the data from the same channel group, as seen in Figure 8. As a result, there is a breakdown in channel communication and a loss of some of the most useful information. The idea behind channel-shuffle is to rearrange the channels in various channel groups such that each group’s information is properly integrated without requiring further calculations, which fixes the issues arising before. This method increases the model’s capacity for learning by avoiding the occurrence where each output feature map is independent of the others. The channel-shuffle implementation technique is given below:

Let’s assume that the input layer is separated into G classes, and each class has N number of channels, then there are (G × N) channels in all. The reshape operation is performed to generate a feature matrix of (G, N) dimensions.

Turn the feature matrix upside down and change its size to (G, N).

Arrange this matrix in rows and columns.

Hu et al. [44] suggested the squeeze-excitation (SE) module in 2018. By determining the significance of feature channels, the self-attention of SE module determines the weight factor of each channel. The learning of feature-channel data with high weight factors is the main emphasis of neural network models. Complex information may be found in PSLs patterns, and pixel noise is more prevalent. Not every channel’s data helps the classifier to make the best choice. To eliminate the duplicate channel information, it is important to examine the significance of each channel using the SE module before channel-switching. This can boost the model’s anti-delay properties, while enhancing the classification precision.

Figure 6a displays the construction of the SE module. The Squeeze and Excitation sections make up the SE module. Through the global pooling layer, the squeeze component turns the input data from W × H × C into 1 × 1 × C data. The characteristic length, breadth, and channel number, respectively, are denoted by W × H, and C. The Excitation part’s primary structure is made up of two fully connected layers (FC). To lower the amount of calculation, the first fully-connected (FC) layer is decreasing the channels. To increase the channels, the second FC layer is utilized. Afterwards, we used the GELU function to regulate the weight factor value between the first FC layer and the second FC layer. Also, we used the sigmoid function to regulate the weight factor value between 0 and 1 FC layer, which are decreasing the channels. The module will then execute a scale operation. To acquire the feature data with re-calibrated weights, multiply the input W × H × C data with the 1 × 1 × C weighting factor generated by Excitation in the appropriate manner.

#### 4.2.4. Transfer Learning

Transfer learning (TL) [45] enables the transfer of information acquired in other activities or domains, which can save training time and enhance performance. Domain, D, is the topic of learning in TL. The source domain (DS) and the destination domain (DT) make up the domain. D = X, P(X), where X is the data and P(X) is the probability distribution that generates X. Task T, the learning objective, is divided into source task Ts and target task Tt. The task may be written as T = {Y, f(.)}, and consists of the label space Y and the prediction function f(.). Given the source domain Ds and source task Ts, the target domain Dt and target task Tt are known. When Ds ≠ Dt or Ts ≠ Tt, transfer learning uses the information acquired in the source domain Ds and the source task Ts to complete the target task Tt in the target domain Dt. TL is implemented in this study using the weights of Inception-v3 on the “ImageNet” dataset.

## 5. Results

### 5.1. Experimental Setup

In experiments, the data are divided into two groups by using a 10-fold cross-validation test for the training set and the test set. The test set is utilized for model evaluation and prediction. Furthermore, we evaluate the classification accuracy of each PSL’s images after cropping (256 × 256 × 3). To ensure fair comparisons, the same settings are used for all models. Additionally, the output layer, which is equal to the number of classes, has taken the place of the model’s final layer. In all networks, the hyperparameters were standardized. “Stochastic Gradient Descent (SGD)” is used to train all network models because it runs quickly and converges well. Due to GPU memory limitations, we train the networks in 64-batch increments. For all networks, the initial learning rate is set to 0.001 and the learning policy is “step” with a gamma of 0.5. Also, we used a set of optimization configurations. By using the “ADAM” optimizer on a categorical cross-entropy loss, the networks were optimized. To perform comparisons, we employ a “momentum” of 0.9 and a weight decay of 5 × 10^4^ All experiments use the branch normalization (BN) technique and the GELU function. To implement and test the ShuffleNet-Light model, we utilized a PyTorch library and a computer Intel Core i7-3770 CPU, 16 GB of RAM with an Nvidia GTX 1080 GPU.

### 5.2. Model Training

Each PSL is down-sampled to produce (256 × 256 × 3) tensors before training the model. Afterward, several CNN pre-training models are then gathered from different sources. The ShuffleNet [11], SqueezeNet [12], ResNet18 [13], MobileNet [14], Inception-v3 [15], Xception [16], and AlexNet [17] are the pretrained TL architectures. The ImageNet dataset was used to train all these TL models, as well as our improved ShuffleNet-Light architecture on 28,000 PSLs. Due to memory limitations, we employ the stochastic gradient descent (SGD) algorithm to optimize the parameters of these pretrained TL models. For Inception-v3 or ShuffleNet-Light models, the learning rate and batch size are 0.001 and 64, respectively. The learning rate and batch size for other networks are 0.01 and 64, respectively. To enhance PSLs classification, the network parameters of these pretrained models must be adjusted and fixed with training samples.

### 5.3. Model Evalaution

To compare the performance of suggested ShuffleNet-Light classifier compared to state-of-the-art solutions, we utilized various statistical metrics. Several metrics, including accuracy (ACC), recall (RL), specificity (SP), precision (PR), F1-score, and Matthew’s correlation coefficient (MCC), have been used in the past. Compared to other metrics, the MCC uses a contingency matrix approach to calculate the “Pearson product-moment correlation coefficient” between actual and expected values. The MCC metric [46] is unaffected by the problem with imbalanced datasets. In this study, these metrics were employed. The True Positive (TP) and True Negative (TN) values are measured, which demonstrate whether the model successfully predicted the data was true or incorrect. “False Positive (FP)” and “False Negative (FN)” statistics show that the algorithm is not correctly recognized the data. In other words, it provides a technique for checking how well the method recognizes the data. These statistical indicators are computed in the manner shown below:(3)Accuracy= (TP+TN)/(TP+TN+FP+FN)×100
(4)Recall=TP/(TP+FN)×100
(5)Specificity=TN/(TN+FP)×100
(6)F1−Score=2×(precision×recall)/(precision+recall)
(7)$$\begin{array}{ll} Matthews\;correlation\; & coefficient\;\left(MCC\right)\\ & =(TP\times TN\\ & -FP\times FN)\\ & /\left(\sqrt{\left(TP+FP\right)\times \left(TP+FN\right)\times \left(TN+FP\right)\times \left(TN+FN\right)}\right) \end{array}$$

### 5.4. Results Analyis

We have selected various architectures of a CNN-based model to compare the performance with the proposed Light-Dermo system. Figure 9 presents representations of upgraded ShuffleNet-Light’s significant feature maps to help comprehend the learning capacity of the channelwise method. It shows how we said that the Light-Dermo model could keep more of an image’s details because it reuses features. Furthermore, as the network grows, the features become increasingly abstract. Based on the best splits of PSL’s dataset with 40 epochs, the Light-Dermo model’s training and testing loss vs. accuracy diagram, along with the AUC curve, are shown in Figure 10. In addition, the Light-Dermo used seven classes of PSLs to recognize actinic keratoses (AK), basal cell carcinoma (BCC), benign keratosis (BK), dermatofibroma (DF), melanoma (MEL), melanocytic nevi (NV), and vascular skin lesions (VASC). Table 4 shows the classification results achieved by the Light-Dermo by developing the ShuffleNet-Light model for recognition of the seven classes of PSLs in terms of SE, SP, ACC, PR, F1-score, and MCC.

For the dataset of pigmented skin lesions (PSLs), we assess the performance of the different DL and TL-based models such as ShuffleNet [11], SqueezeNet [12], ResNet18 [13], MobileNet [14], Inception-v3 [15], Xception [16], and AlexNet [17]. These CNN-based architectures with TL capabilities are compared with the proposed ShuffleNet-Light model in terms of performance. According to Table 4, the ACC of 99%, PR of 98.3%, SE of 97.4%, SP of 98.2%, F1-score of 98.1%, and MCC of 99% are achieved by a proposed ShuffleNet-Light model compared to other CNN-based architectures. The classification accuracy (ACC) for AlexNet is 88.9%, MobileNet is 90.8%, ResNet18 is 89.3%, Inception-v3 is 89.4%, Xception is 89.5%, and ShuffleNet is 88.5%. According to this table, MobileNet achieved a higher ACC compared to others. Also, the original ShuffleNet is not performing well when observed in 40 iterations.

Compared to others, the ShuffleNet-Light model is an effective method for classifying pigmented skin lesions (PSLs) with a higher accuracy. Compared to other CNN and TL-based architectures, the original ShuffleNet network has more parameters and FLOPs (shown in Table 5) than the proposed ShuffleNet-Light model. As a result, the enhanced architecture has fewer parameters and converges more quickly than its baseline. The FLOPS are 67.3 million, the number of parameters is 1.9 million, the model size is 9.3 million, and the GPU speed is 0.6 million, as shown in Table 5. As a result, the ShuffleNet-Light model created a new and improved architecture, as described in Section 4.

We run tests on the pigmented skin lesions (PSLs) dataset to assess the performance of various CNN-based models. The accuracy curves for the improved ShuffleNet-Light suggest that it is a lightweight architecture for the classification of seven classes of PSLs. However, the other network architectures occasionally yield worse results, depending on the number of iterations. The network structure of our proposed model is more complex than their backbones because we concatenated the features of the previous layers, which are thought to be the cause of these situations. Figure 11 also displays the classification results of the basic ShuffleNet and ShuffleNet-Light in terms of accuracy, precision, F1-score, and MCC.

These experiments show that an improved ShuffleNet produces classification rates for seven classes of PSLs that are superior to those of standard models. The improved ShuffleNet-Light-based classification accuracy of PSLs is 99%. As a result, the integration of squeeze-and-excitation (SE) blocks provides the best TL-based architecture for developing a lightweight Light-Dermo model, as shown in Table 6. Also, the dense connection protects against the overfitting problem better, especially when learning from small amounts of data. The layers in the light dermis are thicker. As a result of direct connections between all layers, the network has a very deep design. In this paper, speed is a key performance measure in addition to classification accuracy and a significant assessment criterion for specific application situations. In addition, we have performed state-of-the-art comparisons in terms of computational cost and accuracy, which are described in Section 5.4.

### 5.5. Comparisons in Terms of Computational Time

Table 7 compares the model’s performance with some other current models over PSL datasets in terms of the number of epochs, accuracy, f1-score, MCC, and computational time in seconds (S). In addition, we have also compared the Light-Dermo model in terms of computational cost with others such as 9CNN-models [6], AlexNet [8], MobileNet-LSTM [10], FixCaps [11], EfficientNet [12], CNN-Leaky [13], and DCNN [15]. Those results are described in Table 8. The suggested model has been shown to work very well, even though it only has a few trainable parameters and a short calculation time. PSL images have been used to compare accuracy with other cutting-edge techniques, and the results are shown in Table 7. The result of the analysis demonstrates that the proposed model, when compared to other cutting-edge models, has attained high accuracy. Since different datasets (HAM10000, Kaggle, and clinical pictures) are utilized throughout all investigations, the accuracy varies as shown in Table 8. In this table, the execution time is very small compared to others. In addition, Khan et al. [8] used the VGG16 model architecture to reach an accuracy of 80.46% using the HAM10000 dataset, whereas Agrahari et al. [10] and Chaturvedi et al. [11] used the MobileNet model architecture to obtain accuracy rates of 80.81% and 83.10%, respectively. Compared to them, our Light-Dermo model outperforms them in terms of accuracy and computational time.

In another experiment, we compared CPU, GPU, and TPU processing in terms of batch size and computational performance of the proposed ShuffleNet-Light model. In practice, layer-by-layer analysis of the CNN implementation [43] in the CPU, TPU, and GPU was required. To maximize its performance on TPU, the ShuffleNet-Light network should be built with each job being a MISD (multiple instruction, single data) task. When creating instructions, the tasks of the neural network must be prioritized. In fact, the GPU offers increased flexibility and simple programming for small quantities. Due to the execution pattern in wraps and scheduling on simple on-stream multiprocessors, GPUs suit batch sizes for little data better. By maximizing memory reuse, the GPU works well for huge datasets and network models. Weight reuse in fully linked neural networks is lower and, as a result, increased memory traffic results as the model size grows. The GPU’s memory bandwidth enables it to be used for applications that need memory. Due to the additional parallelism capability, GPUs perform better than CPUs when handling large neural networks. The GPU outperforms the CPU for fully linked neural networks, but the TPU excels with huge batch sizes.

Whereas in the case of TPU, we have utilized the array structure, which works better in the case of TPU, on the ShuffleNet-Light architecture with big batches to offer high throughput during training. To fully utilize the matrix and multiply units in the systolic array of the TPU, large batches of data are required. The speedup in the architecture rises as the batch size does. TPU is the best due to the spatial reuse properties of the networks for large batch sizes and complicated CNNs. The performance of CPU, TPU, and GPU benchmarks in terms of batch size is mentioned in Table 9 for the proposed ShuffleNet-Light model.

We suggested a lightweight CNN and applied the channelwise attention (CA) mechanism on ShuffleNet blocks, which were tested on a huge PSLs dataset for the classification of multiclass to produce cutting-edge results. The top-1 accuracy of ShuffleNet-Light increased to 8.7% when SE modules were combined with the backbone ShuffleNets, for example (shown in Table 9). We discover that ShuffleNets with CA modules are often 25–40% faster than the basic ShuffleNets on mobile devices, despite a minimal decrease in theoretical complexity. This suggests that real-time speedup evaluation is crucial for low-cost architecture design.

The ShuffleNet model gathers the real-time control needs and boosts accuracy, which is the underlying improvement. In addition, the ShuffleNet-Light model further comprehends the learning capacity of the channelwise method. To show this point, the visualizations with numerous major feature maps of enhanced ShuffleNet are provided in Figure 12 and Figure 13, where our suggested model can preserve more image information due to important feature reuse. Furthermore, as the network grows, so do the featured visuals. For the unseen test datasets, it obtains an accuracy of 99.1% and an equivalent score for weighted precision and recall, an f1-score of 98.5%, and MCC of 99%. We have shown that combining two or more models using different assembly approaches can increase a classification model’s capacity for prediction and generalization. By contrasting the true label and predicted label for each item in the test set, the confusion matrix also provides a crystal-clear demonstration, as shown in Figure 12. Even if many of the photos are properly classified, it is hard to achieve high classification capacity for each class due to the high interclass similarity and intraclass variability across images in some classes. The comparison procedure shows that the suggested technique performs better in terms of accuracy, precision, and the f1-score. However, our model is a lot more accurate than this one, so the study that was suggested has a higher MCC.

Figure 13 shows the classification results of the proposed ShuffleNet-Light model, when Xception-v3 is deployed as a pretrained model with data augmentation, consisting of 28,000 PSLs. Whereas in Figure 14, the best classification results of the proposed ShuffleNet-Light model are shown when Xception-V3 is deployed as a pretrained model without data augmentation techniques. Therefore, the proposed ShuffleNet-Light model provides the best classification accuracy when combined with data augmentation techniques. These results also indicate that the proposed ShuffleNet-Light model will be built by using any pretrained model without affecting its performance.

The results show that, regardless of complexity, ShuffleNet models outperform MobileNet or AlexNet architectures. Even though the ShuffleNet network is built for tiny models (150 MFLOPs), it still performs better than MobileNet in terms of computation cost. ShuffleNet performs 6.7% better than MobileNet on networks with fewer nodes (40 MFLOPs). ShuffleNets design features 50 levels (or 44 layers) from a configuration standpoint, whereas MobileNet only has 28 layers.

Table 10 compares the performance of the ShuffleNet-Light architecture with various pretrained TL models such as Inception-v3, AlexNet, and MobileNet. This performance is measured on an Android mobile device with an ARM platform. The real inference speed of ShuffleNet models will be tested as part of the final evaluation. The resulting inference time on the mobile device is displayed in Table 10. The outcome demonstrates that ShuffleNet is significantly quicker than earlier AlexNet-level models or speedup techniques. The ShuffleNet-Light architecture becomes more popular for mobile devices with comparable computing costs and greater performance. ShuffleNet is clearly better than other current models, as shown by the studies on pointwise group convolution and channel shuffle operations, where ShuffleNet performs best, and comparisons with other structural units, where ShuffleNet beats other units by an average of 5%. According to all appearances, there is no noticeable difference when ShuffleNet is implemented using any pre-trained model, but the Inception-v3 model is the best on actual mobile devices. Nevertheless, the cost of calculation has significantly decreased from AlexNet’s 720 of FLOPs to ShuffleNets 140 of FLOPs and 0.5x’s 40 of FLOPs. Future research may decide to concentrate on optimization and performance evaluations on various mobile devices. Given the amount of work that Apple and Google invest in their mobile platforms, it could be useful to compare performance or computation costs between various companies.

Table 11 is used to display that the outcome demonstrates a trend where the PSLs’ recognition scores rise. The ShuffleNets’ decreased categorization error demonstrates the value of cross-group information exchange even better. The lower the classification error, the better ShuffleNet with channel shuffle is in various architectures than ShuffleNet without channel shuffle.

On the HAM10000, ISIC-2019, and ISIC-2020 datasets, a thorough comparison of testing accuracies is also performed by using various optimizers and various learning rates. Based on the number of parameters, Table 12 shows the results of several optimizers. In comparison to other optimizers, SGD and Adam perform exceptionally well, with an accuracy of approximately 99%. Moreover, Adam took somewhat longer (0.0068) than the SGD and Adadelta optimizers, but it had the lowest 0.001 learning rate and 6.50 validation loss. As a result, we have selected the SGD algorithm for the optimization of the ShuffleNet-Light architecture.

### 5.6. Generalized Model: No Overfitting or Underfitting

When a TL model overfits, it tries to fit all the training data and ends up remembering the patterns in the data as well as the noise. The goal of these TL models is defeated since they do not generalize effectively and perform well in the presence of unseen data circumstances. Low bias and high variance are two signs of a model that has been adjusted too much. The training data samples in our suggested ShuffleNet-Light model were sufficient and the training data were cleaned, which aided the model’s ability to generalize its learning capabilities. The model was trained with enough data for many epochs and had a low standard deviation. Underfitting, on the other hand, happens when the TL model is unable to clearly translate the input to the desired variable. When characteristics are not seen completely, the inaccuracy in the training and unseen data samples increases. When the model is unable to learn from the training data and the training error is sufficiently large, it can be identified. Low variance and high bias are the most typical signs of underfitting. With our model, there was no chance of using training data that had not been cleansed, and the dataset was also not changed through augmentation. We can conclude that there was no over- or underfitting in our proposed model.

We conduct a preliminary experiment to determine which activation function has the most impact on model correctness. According to the information in Table 13, the Sigmoid, Tanh, ReLU, GELU, and Leaky-ReLU have accuracies of 96%, 96%, 97%, 96%, and 99%, respectively. GELU, which is around 2% more accurate than the original activation function ReLU, achieves the greatest prediction among these activation functions. As a result, the GELU function serves as the activation function in our improved ShuffleNet model.

### 5.7. Model Interpretability

The visualization impact of the sample with the greatest prediction probability for each category among the test set samples is displayed in Figure 15 to confirm the interpretability and explainability of the ShuffleNet-Light network based on the classification layer described in this article. In this study, the GradCAM score [47] was calculated on seven classes of PSLs. This is the prediction probability value of the test model’s final output in each class of PSLs. Each row in Figure 15 demonstrates that the images are all derived from the seven sample classes of PSLs. On a decent heat map, the GradCAM score can show the lesion areas of pigmented skin lesions. The visual aspect of the image can demonstrate that the regions the model focused on are comparable to those produced by human experience after the image has been examined by qualified physicians. The visualization effect of ShuffleNet-Light, which is superior, demonstrates that the localization area is small and that all results are contained in the lesion area. This demonstrates that the proposed model is more interpretable and explicable in seven PSLs classes.

## 6. Discussion

The last ten years have seen the emergence of several novel yet superior computer-aided diagnosis (CAD) systems to classify pigmented skin lesions (PSLs). According to studies [8], a significant portion of skin cancers may be very well prevented. Early detection of abnormal changes on the skin can aid in the diagnosis of skin cancer and stop it from spreading to other organs. The PSLs’ skin lesions should be diagnosed and graded as soon as possible to protect the patient’s health. Melanoma can infiltrate deeply into the skin if it is detected too late. As the condition advances, treatment becomes more challenging. Digital dermatoscopy is being employed for the early detection of PSLs, but even for dermatologists with experience, it remains a challenging process. Patients have additional challenges since they must make repeated trips to the doctor to track and detect variations in skin color [22]. The patient’s life is in danger since this treatment is laborious and prone to mistakes. In the past, as mentioned in Table 1, several authors developed two-class-based (melanoma versus nevus) classification systems compared to multiclass PSL lesions. The majority of the studies required extensive image processing domain knowledge and hand-crafted feature extraction, which limits detection accuracy, compromises generalizability, and is computationally expensive.

As a result, a quicker and more precise way of identifying and categorizing skin cancer is required [9]. The literature has mentioned a variety of methods for detecting cancer from comparatively small datasets [2,5,9]. However, a thorough investigation of the effect on a sizable database is still pending. Many classification techniques rely significantly on manually created feature sets, which in dermatoscopic skin pictures have a limited capacity for generalization. Lesions exhibit a high degree of visual likeness and are highly connected with one another due to their closeness in color, shape, and size, which leads to poor feature information. Hand-crafted feature-based skin categorization methods are therefore useless [5,9,10]. In this situation, a deep learning (DL)-based approach has proven to be effective. Traditional machine learning (ML) techniques require handcrafted features, while deep learning (DL) systems have a large ability to extract complex, detailed, task-specific, and effective features, enabling them to develop an elegant model with better performance [3,11]. Dermatoscopic analysis may be minimized, and DL-based skin cancer detection is economical. Thus, using the approach outlined above, skin cancer disorders can be identified early enough.

We suggest innovative designs that pair the channelwise attention (CA) mechanism with effective pretrained CNN networks to adhere to resource constraints (latency, memory size, etc.). We have selected ShuffleNet [5] as the backbones because of its effective topologies, which not only provide tiny networks but also allow for the encoding of more data. Squeeze-and-excitation (SE) blocks, which are regarded as CA mechanisms, are combined with backbones to further increase the precision of pigmented skin lesions (PSLs) categorization. Comparative trials for the categorization of seven categories of PSLs demonstrate that this proposed lightweight CNN (Light-Dermo) model is more successful than its alternative structures. This outcome demonstrates that accuracy may be greatly increased. Finally, the Light-Dermo can handle real-time applications. The Light-Dermo can assist in reducing overfitting. The ShuffleNet network’s connections created quick pathways from the bottom layers to the top ones. As a result, the loss function (GELU) gives each layer more direction. Therefore, the dense connection protects against the overfitting problem better, especially when learning from small amounts of data. The layers in the light dermis are thicker. As a result of direct connections between all layers, the network has a very deep design. Multiple PSL lesions are greatly imbalanced in the HAM10000, ISIC-2019, and ISIC-2000 datasets. To address this issue, powerful data augmentation techniques are applied. We used cross-validation to test our model, and the results showed that it worked much better than the existing methods. The Light-Dermo is a computationally inexpensive solution for the diagnosis of pigmented skin lesions (PSLs) compared to state-of-the-art approaches.

The visualizations with numerous major feature maps of enhanced ShuffleNet are provided to further understand the learning capacity of the channelwise method, as shown in Figure 8 and Figure 9. The most recent ISIC-2019, ISIC-2020, and HAM10000 datasets are used in this paper, which show cutting-edge dermoscopy images of the most recent developments in cancer lesion identification, where our suggested model can preserve more image information due to important feature reuse. Furthermore, as the network grows, so do the featured visuals. For the unseen test datasets, it obtains an accuracy of 99.1% and an equivalent score for weighted precision, recall, and f1-score of 98.5%. Several visual examples of the performance of the proposed Light-Dermo model are displayed in Figure 10, Figure 11 and Figure 12. The significant results were attributed to squeeze-and-excitation (SE) blocks, which were integrated into the ShuffleNet architecture for the development of a lightweight Light-Dermo model. The Light-Dermo can assist in reducing overfitting. In patients undergoing dermatoscopy who may have a pathogenic infection, this research proposes a novel, highly effective, and accurate DL model for quick and non-invasive detection of skin cancer. A visual example of the proposed ShuffleNet-Light model is shown in Figure 16.

Due to memory access and other overheads, we find that for every four percent reduction in theoretical complexity, our implementation typically speeds up by 2.6 percent. Even though the theoretical speedup is 1.8%, our ShuffleNet 0.5 model still gets a 10-fold real speedup compared to Inception-v3, AlexNet, and MobileNet, which is much faster than earlier AlexNet-level models or speedup methods. The resulting inference time on the mobile device is also displayed in Table 8. The outcome demonstrates that ShuffleNet is significantly quicker than earlier AlexNet-level models or speedup techniques. The next phase of artificial intelligence will involve implementing neural networks on mobile devices, and ShuffleNet is very effectively described in this study. The ShuffleNet will become more popular in the CNN space for mobile devices with comparable computing costs and greater performance. ShuffleNet is clearly better than other current models, as shown by the studies on pointwise group convolution and channel shuffle operations, where ShuffleNet performs best, and comparisons with other structural units, where ShuffleNet beats other units by an average of 5%. According to all appearances, there is no discernible difference when ShuffleNet is implemented using any pre-trained model, but the Inception-v3 model is the best on actual mobile devices.

Several TL-based pre-trained models can be used to classify multiclass PSL lesions. The ShuffleNet [11], SqueezeNet [12], ResNet18 [13], MobileNet [14], Inception-v3 [15], Xception [16], and AlexNet [17] are the TL architectures. For the development of an improved ShuffleNet-Light model, we used a better activation function (GELU) and added an attention mechanism. According to our studies, each of these models can perform multiclass recognition of PSLs. However, the ShuffleNet serves as the primary baseline and backbone model. Additional comparisons were also made between the Inception-v3 and ResNet-18 models. With the information provided above, we recommend using the attention mechanism SE block as a potential optimization technique. The model was further improved by adding the activation function. Finally, the GELU performs better than the ReLU. The models’ final F1 scores (accuracy, precision, and recall) and MCC were determined. Out of the deep learning models chosen for this investigation, the optimized ShuffleNet-Light performed the best. In addition, our improved ShuffleNet-Light model outperformed previously published techniques in terms of multiclass recognition of PSL lesions.

### 6.1. Advantages of Proposed Approach

To get correct results, data imbalance is an essential stage in classification tasks. To accurately identify images, deep learning (DL) algorithms employ multiple layers of artificial neurons. However, the class imbalance has been adjusted by using the data augmentation technique, which was applied to (256 × 256 × 3) pixels. If this step is not performed on datasets, then more memory and processing time are needed. Deep learning (DL) or machine learning (ML) algorithms are often overfitted due to their convoluted design. Our job is to provide a less complicated structure and reduce computational time. Our ShuffleNet-Light architecture with a balanced layer architecture is what we then suggest. We added various blocks to the architecture with kernel regularizes of 0.001 to improve performance. The main function of kernel regularization is to address overfitting problems. Due to the multiclass classification problem, we applied a softmax classifier in the study to recognize seven classes of PSLs. Consequently, all types of optimizers function equally well, except GELU for us, and our model generates results quickly based on it. Three well-known, publicly accessible datasets are used in the study. On these datasets, the suggested deep ShuffleNet-Light DL model outperforms others. In short, the following advantages of the proposed study are described below:(1)The most important thing that our study adds is the idea of a new, highly optimized, lightweight CNN model that can find seven types of pigmented skin lesions (PSLs).(2)The ShuffleNet-Light design minimizes network complexity and increases accuracy and recognition speed compared to those of currently available deep learning (DL) architectures. Our model’s ShuffleNet architecture’s SE block adds to its accuracy by modifying it. It also doesn’t have much of an impact on the model’s complexity or recognition rate.(3)The ShuffleNet-Light has a generalized capability with no overfitting or underfitting problems.(4)Since the activation function improves accuracy, we replaced the original ReLU function in the proposed model with the GELU function, which, in our analysis, improved the model. Therefore, the activation function does influence how well the model does when it must classify skin lesions.(5)We think that our research shows that it might be possible to recognize and track PSLs in real-time when mobile phones are used in outreach settings.

### 6.2. Limitations of Proposed Approach and Future Works

Different kinds of pigmented skin lesions (PSLs) are possible. There are only seven different kinds of PSLs included in this study. There are another nine classes of PSLs that will be included in the future to check the performance of the proposed TL-based model. In addition, it is well known that every DL-based technique needs a lot of data to effectively train the model. On the other hand, there are not enough images utilized in the study to train the suggested model.

In this study, we improved the ShuffleNet model with a channel attention (CA) mechanism that achieved acceptable accuracy when compared with other recognition models. Although we performed well in terms of the model’s complexity and recognition speed, its accuracy might still use some work. One of the things we discovered was that the size of the dataset collection we utilized was still insufficient. Furthermore, the nine-class data of PSLs is not examined by our approach. Regarding the first problem, there are not enough datasets and research papers on the recognition of multiple classes of PSLs. Unbalanced sample distribution is a problem that affects all datasets. However, we have performed a data augmentation technique to balance it. Still, we need a balanced dataset. With the available dataset, nothing can really be done to change this scenario. A future study may concentrate on evaluating our suggested model with bigger and more representative PSLs. Regarding the second concern, it should be noted that when optimizing ShuffleNet, we considered the fact that enhancing the analysis of timing data will certainly increase the amount of calculation needed for the model, undermining our desire for it to be lightweight. Other possible research areas include increasing the dataset’s size and applying lightweight models to PSLs data analysis. ShuffleNet-Light is a thin neural network, so further research based on tiny devices, such as tablets and handheld devices with GPU processors, may be taken into consideration to investigate the viability of developing miniature engagement detection devices.

In addition, a ShuffleNet-Light model based on a pretrained technique is proposed in this paper, which has become a dominant idea in recent years. The development model is computationally efficient and effective for deep feature classification of multiclass PSL lesions. Recently, a graph-based approach [48] is also utilized to extract representative image features. However, it is necessary to compare this model with a graph-based technique to see the comparisons in terms of accuracy and computational efficiency. This step will be addressed in the future.

## 7. Conclusions

This study suggests an enhanced TL-based pretrained model based on the ShuffleNet architecture. To reduce computational complexity, we adopted the “ShuffleNet v2” model. The model’s activation function was subsequently enhanced when we included an attention mechanism utilizing SE blocks. Various pre-trained CNN models were chosen to assess and contrast the performances. Accuracy, F1-score, MCC, FLOPs, and speed were utilized to evaluate these models’ performances. According to the outcomes of our test, our improved ShuffleNet-Light model performed the best. When compared to other published models or methodologies utilizing the same dataset, our model had the greatest accuracy. The improved ShuffleNet-Light model is suited for mobile platforms since it is a lightweight framework. Conventional neural networks are still used in the automatic diagnosis of pigmented skin diseases, despite lightweight networks’ widespread use in other industries. In this research, a novel architecture (ShuffleNet-Light) has been suggested for the categorization of seven classes of PSLs. Several state-of-the-art CNN-based TL models are tested in our studies. Results reveal that ShuffleNet obtains the maximum accuracy, which reached 99.01 percent. When it comes to complexity, Inception-v3 performs the best. The ShuffleNet-Light is better suited to our task when weighing complexity and accuracy. As a result, we incorporate ShuffleNet into our network design. Experimental findings show that, when compared to the baseline, our proposed model has better discrimination capability and classification accuracy in multiclass recognition of PSLs.

## Figures and Tables

**Figure 1 diagnostics-13-00385-f001:**
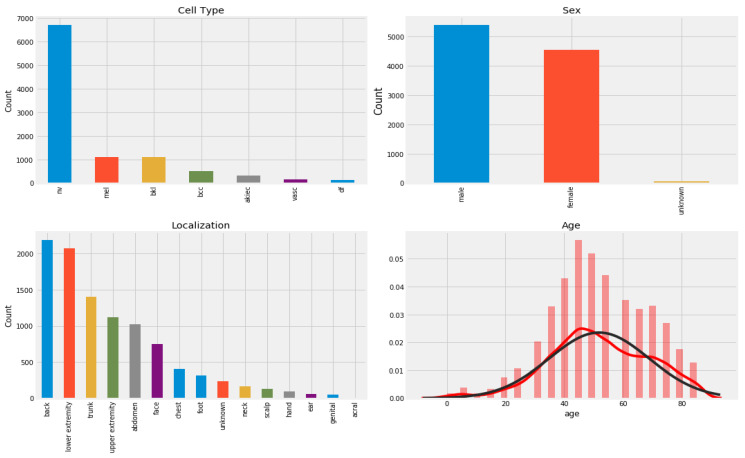
Distribution of sample in HAM10000 dataset according to patients’ history and type of pigmented skin lesions (PSLs).

**Figure 2 diagnostics-13-00385-f002:**
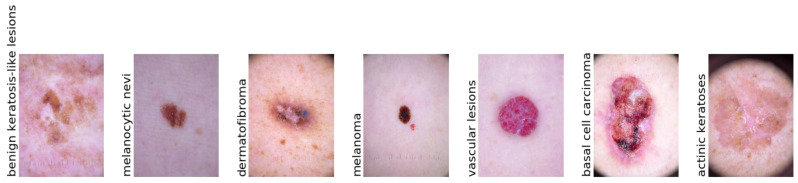
A visual example of a selection of region-of -interests (ROIs) from PSLs lesions after automatically selecting from centroid position.

**Figure 3 diagnostics-13-00385-f003:**
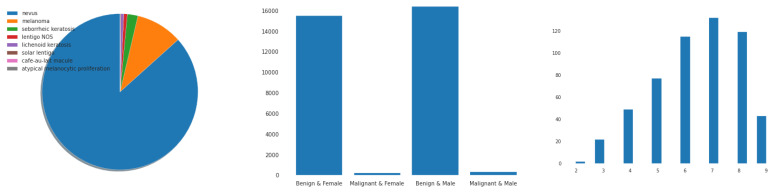
Distribution of samples in the ISIC-2019 and ISIC-2020 datasets according to patients’ history and type of pigmented skin lesions (PSLs).

**Figure 4 diagnostics-13-00385-f004:**
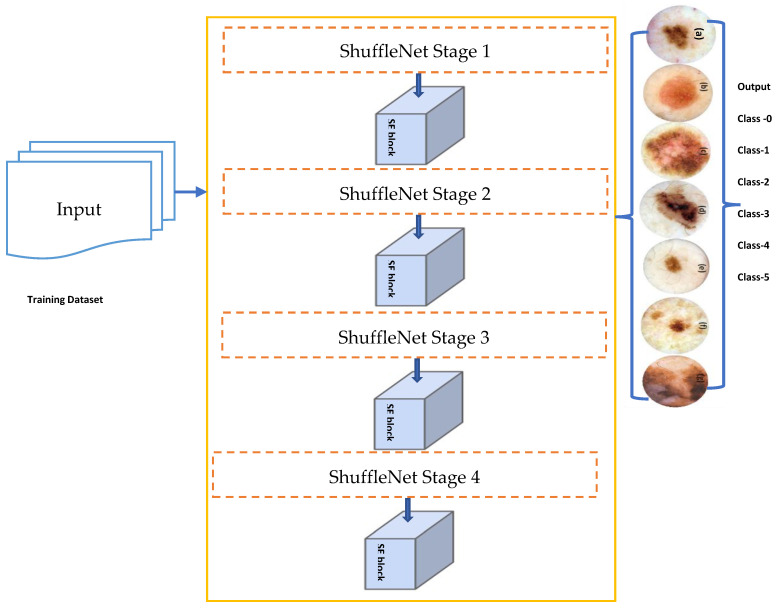
A systematic flow diagram of the improved ShuffleNet architecture.

**Figure 5 diagnostics-13-00385-f005:**
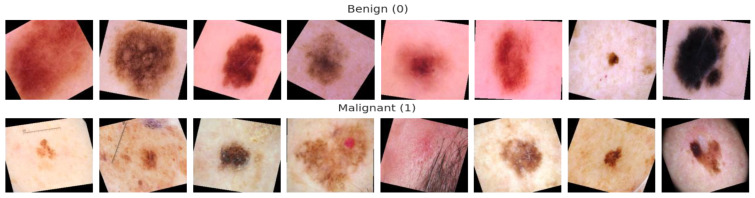
An example of data augmentation techniques applied on the selected datasets from the ISIC-2019, ISIC-2020, and HAM10000 sources in case of benign and malignant skin lesions.

**Figure 6 diagnostics-13-00385-f006:**
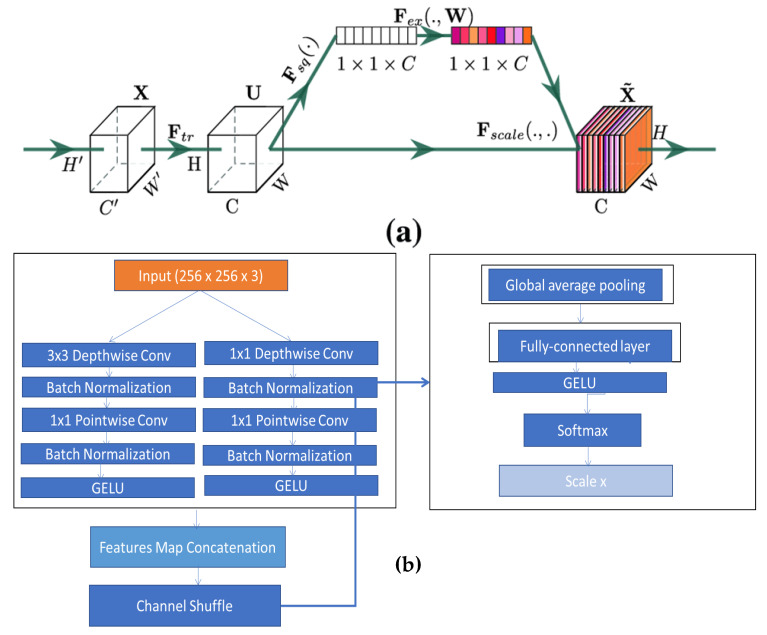
The modules of improved ShuffleNet by using depthwise and pointwise separable CNN along with channel shuffle operations, where (**a**) squeeze and excitation (SE) block, and (**b**) shows the proposed ShuffleNet-Light architecture.

**Figure 7 diagnostics-13-00385-f007:**
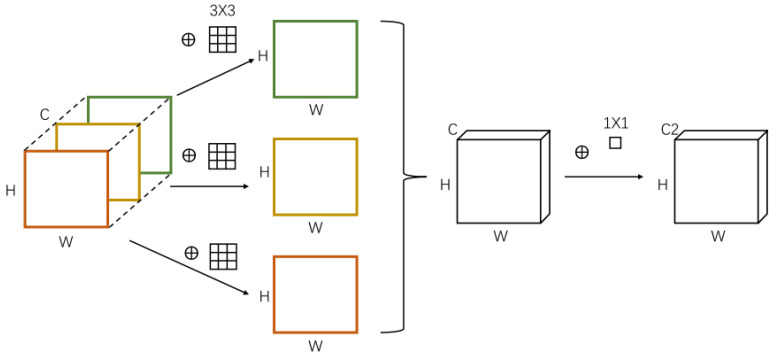
A structure of depthwise separable convolution neural network.

**Figure 8 diagnostics-13-00385-f008:**
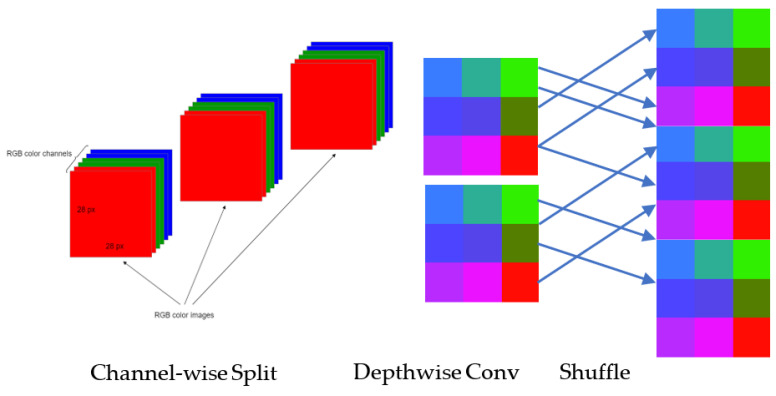
A visual demonstration of the channel-wise shuffle operation.

**Figure 9 diagnostics-13-00385-f009:**
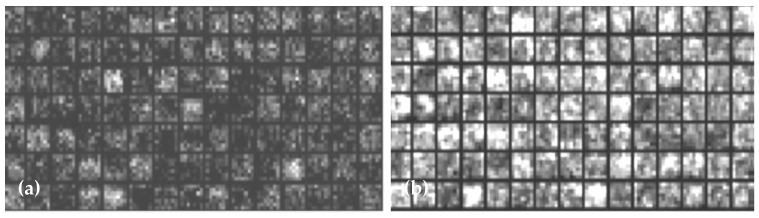
Visualization of images of the networks based on the two versions of ShuffleNet model, where (**a**) shows the seven layers through original ShuffleNet model and (**b**) displays the proposed ShuffleNet model.

**Figure 10 diagnostics-13-00385-f010:**
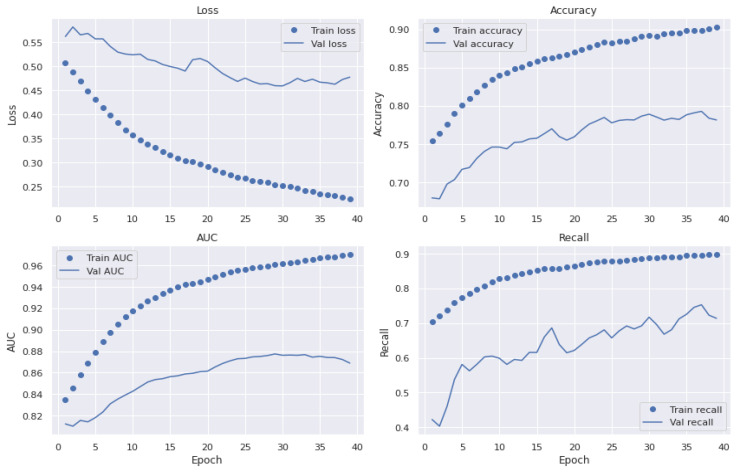
An example of progress improvements in a training and validation data set in terms of accuracy, AUC, and recall by the improved ShuffleNet learning.

**Figure 11 diagnostics-13-00385-f011:**
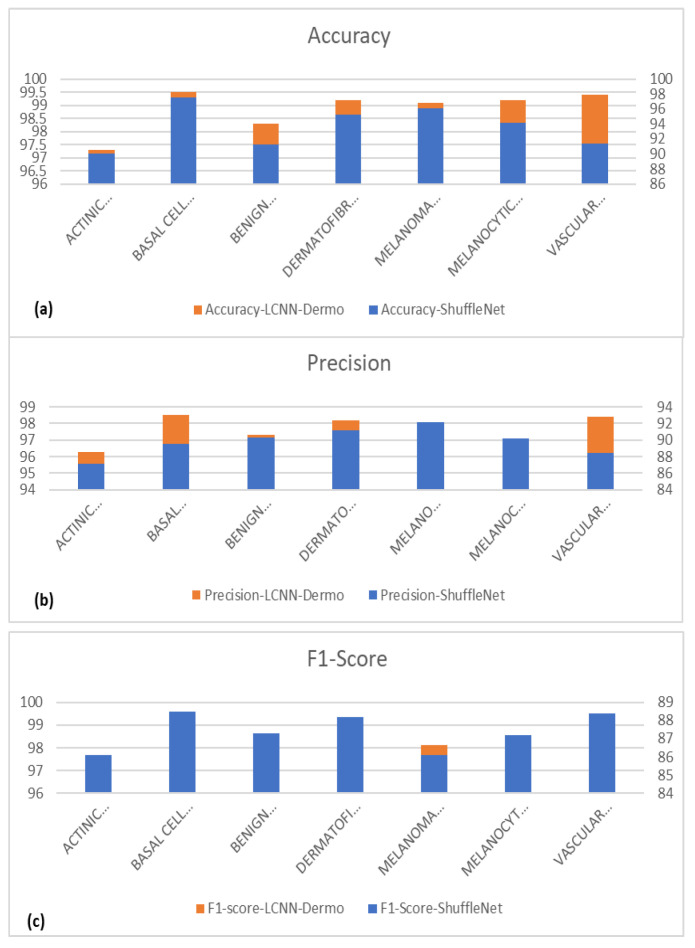
Classification results of the basic ShuffleNet and ShuffleNet-Light in terms of accuracy, precision, and F1-score.

**Figure 12 diagnostics-13-00385-f012:**
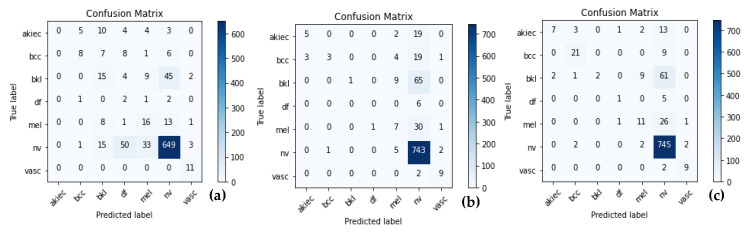
Comparisons of various state-of-the-art systems, where subfigure (**a**) shows the AlexNet [8], (**b**) shows the MobileNet-LSTM [10], and (**c**) the EfficientNet [12] models.

**Figure 13 diagnostics-13-00385-f013:**
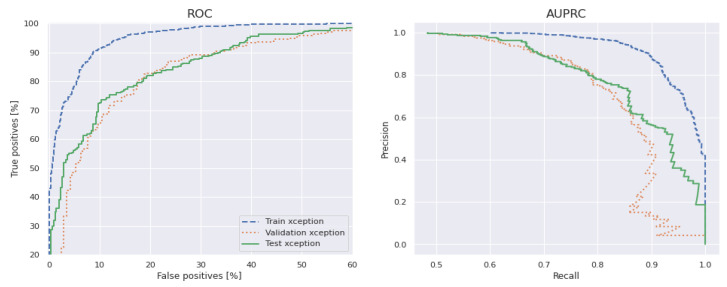
Classification results of the proposed ShuffleNet-Light, when Xception-v3 is deployed as a pretrained model with data augmentation.

**Figure 14 diagnostics-13-00385-f014:**
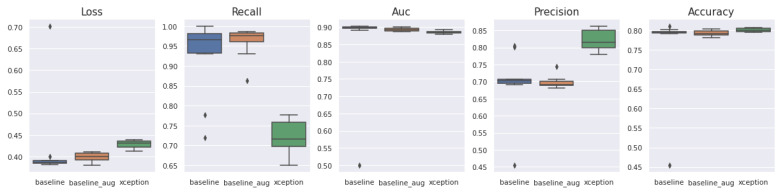
Classification results of the proposed ShuffleNet-Light, when Xception-v3 is deployed as a pretrained model without data augmentation techniques.

**Figure 15 diagnostics-13-00385-f015:**
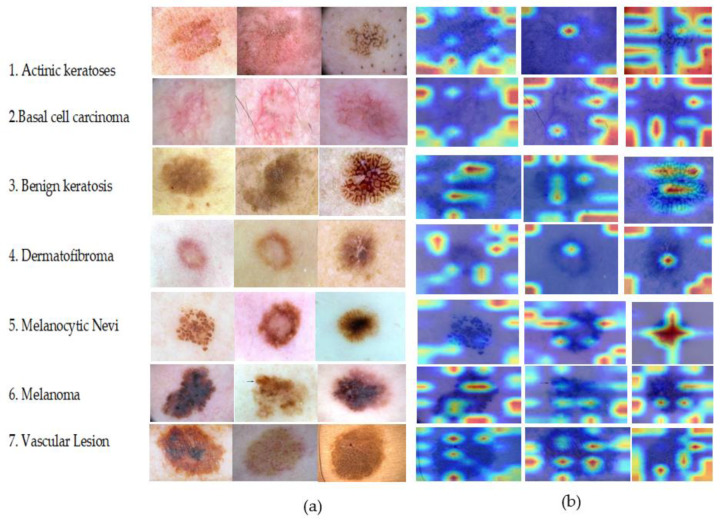
A visual impact of classification results of the proposed ShuffleNet-Light model by GradCAM with the greatest prediction probability for each category among the test set samples. In this figure, (**a**) is the original input sample, and (**b**) is the GradCAM map.

**Figure 16 diagnostics-13-00385-f016:**
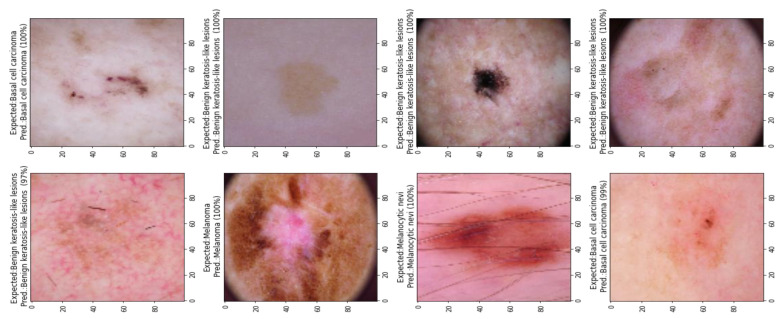
A visual example of classification results of the proposed ShuffleNet-Light model in terms.

**Table 2 diagnostics-13-00385-t002:** Number of images in each class, which were selected from the HAM10000, ISIC-2019, and ISIC-2020 datasets.

Classes ^*^	No. of Images	Data Augmentation
AK	500	4000
BCC	2000	4000
BK	2000	4000
DF	200	4000
NV	6000	4000
MEL	3000	4000
VASC	300	4000
Total	14,000	2800

* Actinic keratoses (AK), basal cell carcinoma (BCC), benign keratosis (BK), dermatofibroma (DF), melanoma (MEL), melanocytic nevi (NV), and vascular skin lesions (VASC).

**Table 3 diagnostics-13-00385-t003:** Data augmentation techniques used to develop a Light-Dermo system.

Parameters	Angle	Brightness	Zoom	Shear	Mode	Horizontal	Vertical	Rescale	Noise
values	30°	[0.9, 1.1]	0.1	0.1	Constant	Flip	Flip	1./255	0.45

**Table 4 diagnostics-13-00385-t004:** Classification results achieve by the proposed ShuffleNet-Light for recognition of the seven classes of PSLs.

Classes	^1^ ACC	^1^ PR	^1^ SE	^1^ SP	^1^ F1-Score	^1^ MCC
(1) ACTINIC KERATOSES (AKIEC)	96.1 ± 1.3	94.2 ± 2.0	96.2 ± 1.4	94.0 ± 2.2	95.2 ± 1.2	97.4
(2) BASAL CELL CARCINOMA (BCC)	97.5 ± 0.8	96.1 ± 1.4	97.0 ± 0.2	95.9 ± 1.5	97.5 ± 0.8	98.4
(3) BENIGN KERATOSIS (BKL)	98.3 ± 0.8	97.2 ± 1.2	99.6 ± 0.3	97.1 ± 1.4	98.4 ± 0.7	96.0
(4) DERMATOFIBROMA (DF)	99.2 ± 0.3	99.1 ± 0.5	99.4 ± 0.5	99.1 ± 0.5	98.2 ± 0.3	98.0
(5) MELANOMA (MEL)	98.1 ± 0.5	99.0 ± 0.5	98.3 ± 0.5	98.9 ± 0.6	99.1 ± 0.5	100.0
(6) MELANOCYTIC NEVI (NV)	98.2 ± 0.3	99.2 ± 0.4	97.3 ± 0.5	99.2 ± 0.4	98.2 ± 0.3	96.0
(7) VASCULAR SKIN LESIONS (VASC)	99.4 ± 0.4	99.6 ± 0.3	99.1 ± 0.6	99.6 ± 0.3	99.4 ± 0.4	99.5

^1^ SE: Sensitivity, SP: Specificity, RL: Recall, PR: Precision, ACC: Accuracy, MCC: Matthew’s correlation coefficient.

**Table 5 diagnostics-13-00385-t005:** Computational performance of the different models.

TL Architectures	Complexity (FLOPs)	# Parameters (M)	Model Size (MB)	GPU Speed (MS)
ShuffleNet-Light	67.3 M	1.9	9.3	0.6
ShuffleNet	98.9 M	2.5	14.5	1.7
SqueezeNet	94.4 M	2.4	12.3	1.2
ResNet18	275.8 M	2.7	15.2	2.6
MobileNet	285.8 M	3.4	16.3	2.7
Inception-v3	654.3 M	3.9	17.5	2.9
Xception	66.9 M	2.5	14.5	2.7
AlexNet	295.8 M	2.5	12.3	3.3

FLOPs: floating-point operations, M: millions, MB: megabyte, MS: milliseconds.

**Table 6 diagnostics-13-00385-t006:** Comparisons results of the various TL-based DL models.

State-of-the-Art	Classes	Augment	Epochs	^1^ Time (S)	^1^ ACC	F1-Score	^1^ MCC
Light-Dermo Model	7	Yes	40	2.4	98.1%	98.1%	98.1%
9CNN models [6]	7	Yes	40	12	80.5%	80.5%	82.5%
AlexNet [8]	7	Yes	40	17	81.9%	81.9%	81.9%
MobileNet-LSTM [10]	7	Yes	40	13	82.3%	82.3%	80.3%
FixCaps [11]	7	Yes	40	15	84.8%	84.8%	83.8%
EfficientNet [12]	7	Yes	40	18	75.4%	75.4%	74.4%
CNN-Leaky [13]	7	Yes	40	20	76.5%	76.5%	75.5%
DCNN [15]	7	Yes	40	22	77.9%	77.9%	76.9%

^1^ ACC: Accuracy, MCC: Matthew’s correlation coefficient, S: Seconds.

**Table 7 diagnostics-13-00385-t007:** Computational performance of the different TL-based deep learning models.

Deep Learning Models	^1^ ACC	^1^ PR	^1^ SE	^1^ SP	F1-Score	^1^ MCC
ShuffleNet-Light	99.1%	98.3%	97.4%	98.2%	98.1%	98%
ShuffleNet	88.5%	87.3%	88.3%	88.7%	87.7%	85%
SqueezeNet	87.9%	84.5%	90.8%	85.4%	87.6%	84%
ResNet18	89.3%	87.1%	93.1%	87.9%	90.1%	89%
MobileNet	90.8%	90.2%	90.0%	91.4%	90.1%	88.%
Inception-v3	89.4%	87.7%	90.0%	88.9%	88.8%	88%
Xception	89.5%	88.3%	89.3%	90.7%	88.7%	88%
AlexNet [17]	88.9%	87.6%	88.8%	88.9%	87.7%	86%

^1^ SE: Sensitivity, SP: Specificity, RL: Recall, PR: Precision, ACC: Accuracy, MCC: Matthew’s correlation coefficient.

**Table 8 diagnostics-13-00385-t008:** Computational performance of state-of-the-art seven classes. Experiments were performed on three datasets such as the ISIC-2019, ISIC-2020 and HAM10000.

CNN-TL Architectures	Complexity (FLOPs)	# Parameters (M)	Model Size (MB)	GPU Speed (MS)
ShuffleNet-Light	67.3 M	1.9	9.3	0.6
ShuffleNet	98.9 M	2.5	14.5	1.7
SqueezeNet	94.4 M	2.4	12.3	1.2
ResNet18	275.8 M	2.7	15.2	2.6
MobileNet	285.8 M	3.4	16.3	2.7
Inception-v3	654.3 M	3.9	17.5	2.9
Xception	66.9 M	2.5	14.5	2.7
AlexNet	295.8 M	2.5	12.3	3.3

FLOPs: floating-point operations, M: millions, MB: megabyte, MS: milliseconds.

**Table 9 diagnostics-13-00385-t009:** Performance of CPU/TPU/GPU Comparisons of the proposed ShuffleNet-Light model.

Batch Size	Epochs	CPU/TPU/GPU (MS)
64	40	600/400/500
128	40	800/450/550
256	40	900/500/600
512	40	900/500/600
1024	40	900/500/600

MS: milliseconds, CPU: central processing unit, GPU: graphical processing unit, and TPU: Tensor Processing Units.

**Table 10 diagnostics-13-00385-t010:** Performance of the proposed ShuffleNet-Light model on an Android mobile device.

Model	^1^ FLOPs	256 × 256	300 × 300	400 × 400
ShuffleNet	80 M	50.2 ms	54.4 ms	54.4 ms
ShuffleNet-Light+Inception-v3	60 M	34.5 ms	35.9 ms	35.9 ms
ShuffleNet-Light+AlexNet	220 M	40.8 ms	45.4 ms	45.4 ms
ShuffleNet-Light+MobileNet	120 M	47.7 ms	50.7 ms	50.7 ms

ms: milliseconds, ^1^ FLOPs: floating-point operations.

**Table 11 diagnostics-13-00385-t011:** Comparisons result of ShuffleNet models with or without channel shuffle.

Model	Shuffle	No Shuffle	%Err
ShuffleNet	40%	50%	2.3
ShuffleNet-Light+Inception-v3	55%	65%	1.4
ShuffleNet-Light+AlexNet	60%	70%	3.2
ShuffleNet-Light+MobileNet	40%	50%	2.2

Err: classification error.

**Table 12 diagnostics-13-00385-t012:** Performance of the proposed ShuffleNet-Light model on different optimizers.

Model	Adam	RMSprop	SGD	AdaMax	Adadelta	Nadam
ShuffleNet	2.9 m	3.9 m	2.9 m	3.2 m	3.2 m	3.0 m
ShuffleNet-Light+Inception-v3	2.0 m	2.1 m	1.9 m	2.1 m	2.1 m	2.0 m
ShuffleNet-Light+AlexNet	2.4 m	3.4 m	2.4 m	3.4 m	3.4 m	2.5 m
ShuffleNet-Light+MobileNet	2.5 m	3.5 m	2.5 m	4.5 m	3.5 m	2.7 m

m: millions, SGD: stochastic gradient descent, RMSprop: root mean square propagation, Adagrad: adaptive gradient algorithm, Adadelta: an extension of Adagrad, Adam: adaptive moment estimation, AdaMax: a variant of Adam, Nadam: Nesterov-accelerated adaptive moment estimation.

**Table 13 diagnostics-13-00385-t013:** Various activation functions compared in terms of accuracy measure.

Model	Sigmoid	Tanh	ReLU	Leaky-ReLU	GELU
ShuffleNet	89%	89%	88%	92%	90%
ShuffleNet-Light+Inception-v3	96%	96%	97%	96%	99%
ShuffleNet-Light+AlexNet	90%	91%	89%	89%	90%
ShuffleNet-Light+MobileNet	90%	90%	88%	87%	89%

Tanh: hyperbolic tangent function, ReLU: rectified linear unit, Leaky-ReLU: leaky rectified linear unit, GELU: Gaussian error linear unit.

## Data Availability

The dataset of the seven categories of PSLs lesions [30] can be obtained in the repository: https://www.kaggle.com/datasets/kmader/skin-cancer-mnist-ham10000 (accessed on 1 January 2021). ISIC-2019 can be downloaded in the repository: https://www.kaggle.com/datasets/andrewmvd/isic-2019 (accessed on 2 March 2021), and the ISIC-2020 dataset can be obtained from the link: https://www.kaggle.com/c/siim-isic-melanoma-classification (accessed on 1 January 2022).

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
