# Peer review of "Light-Dermo: A Lightweight Pretrained Convolution Neural Network for the Diagnosis of Multiclass Skin Lesions"

_diagnostics, 2023, doi:10.3390/diagnostics13030385_

Round 1

Reviewer 1 Report (New Reviewer)

The authors presented transfer learning-based CNN model for the classification of pigmented skin lesions. The original ShuffleNet architecture is used. The model has an accuracy of 99.14%, specificity of 98.20%, sensitivity of 97.45% and 22 F1-score of 98.1%. The experimental results show that the Light-Dermo assists the dermatologist for better diagnosis of PSLs. Although the manuscript has merit, it needs further review addressing the following comments.

1. The authors need to re-write the contributions. The contributions should be very precise not what they have used.

2. What is the novelty? Seems used some existing methods to develop the CNN model

3. Section 3 discussion is redundant. Good for thesis or research report. Need to discuss the methods that are relevant and considered for the study.

4. The proposed model has four ShuffleNet stage. What is the justification?

5. Why ShuffleNet architecture is used as backbone?

6. The authors need to discuss the model’s parameter for training.

7. How the model is tuned for overfitting or underfitting?

8. First few lines of the Abstract needs re-writing. It is confused.

9. Introduction section is very general. Need to discuss the challenges and motivation of the work.

10. Extensive editing of English language and style is required.

Author Response

Original Manuscript ID diagnostics-2150020      

Original Article Title: Light-Dermo: A Lightweight Pretrained Convolution Neural Networks for Diagnosis of Pigmented Skin Lesions

To: Editor in Chief,

MDPI, Diagnostics

Re: Response to reviewers

Dear Editor,

Many thanks for insightful comments and suggestions of the referees. Thank you for allowing a resubmission of our manuscript, with an opportunity to address the reviewers’ comments.

We are uploading (a) our point-by-point response to the comments (below) (response to reviewers), (b) an updated manuscript with yellow highlighting indicating changes, and (c) a clean updated manuscript without highlights (PDF main document).

By following reviewers’ comments, we made substantial modifications in our paper to improve its clarity and readability. In our revised paper, we represent the improved manuscript.

We have done following modifications in short.

According to our 10 years of experience in Skin cancer, this is a novel study, which has been tested on computational devices as well.

(1) All reviewers’ comments related to abstract, introduction, methodology, contributions, results, comparisons, and discussions and conclusions.

(2) Literature review has been extended by adding new references and literature review with each study limitations.

(3) English writing has been improved with no grammatical mistakes in the paper.

(4) References has been written according to journal style.

We have made the following modifications as desired by the reviewers and editorial board, which are attached to this letter.

Best regards,

Corresponding Author,

Dr. Abdul Rauf Baig (On behalf of the authors),

Dr. Qaisar Abbas

Professor.

Reviewer 2 Report (New Reviewer)

The paper presents an interesting idea but some parts of it should be modified:

- Figure 6 should be better highlighted and described as central;

- Considering the imbalance factor of the data, more sensitive performance measures such as Matthew's correlation coefficient or balanced accuracy should be used;

- In support of paragraph 4.1, a table should be added that describes the datasets in a smart way (for quick reading);

- Results without the balancing step of section 4.2.1 should also be shown in order to highlight its benefits;

- Deep learning is dominant as an idea in recent years. What happens if you use a graph approach in representing images? A recent paper that should be cited shows a features extraction algorithm:

Manipur, I., Manzo, M., Granata, I., Giordano, M., Maddalena, L., & Guarracino, M. R. (2021). Netpro2vec: a graph embedding framework for biomedical applications. IEEE/ACM Transactions on Computational Biology and Bioinformatics19(2), 729-740.

Author Response

Original Manuscript ID diagnostics-2150020      

Original Article Title: Light-Dermo: A Lightweight Pretrained Convolution Neural Networks for Diagnosis of Pigmented Skin Lesions

To: Editor in Chief,

MDPI, Diagnostics

Re: Response to reviewers

Dear Editor,

Many thanks for insightful comments and suggestions of the referees. Thank you for allowing a resubmission of our manuscript, with an opportunity to address the reviewers’ comments.

We are uploading (a) our point-by-point response to the comments (below) (response to reviewers), (b) an updated manuscript with yellow highlighting indicating changes, and (c) a clean updated manuscript without highlights (PDF main document).

By following reviewers’ comments, we made substantial modifications in our paper to improve its clarity and readability. In our revised paper, we represent the improved manuscript.

We have done following modifications in short.

According to our 10 years of experience in Skin cancer, this is a novel study, which has been tested on computational devices as well.

(1) All reviewers’ comments related to abstract, introduction, methodology, contributions, results, comparisons, and discussions and conclusions.

(2) Literature review has been extended by adding new references and literature review with each study limitations.

(3) English writing has been improved with no grammatical mistakes in the paper.

(4) References has been written according to journal style.

We have made the following modifications as desired by the reviewers and editorial board, which are attached to this letter.

Best regards,

Corresponding Author,

Dr. Abdul Rauf Baig (On behalf of the authors),

Dr. Qaisar Abbas

Professor.

Round 2

Reviewer 1 Report (New Reviewer)

Better to revise the listed contributions. Some are not really contributions but the method used.

I appreciate the authors for addressing the comments.

Good luck!!!

Author Response

Original Manuscript ID diagnostics-2150020      

Original Article Title: Light-Dermo: A Lightweight Pretrained Convolution Neural Networks for Diagnosis of Pigmented Skin Lesions

To: Editor in Chief,

MDPI, Diagnostics

Re: Response to reviewers

Dear Editor,

Many thanks for insightful comments and suggestions of the referees. Thank you for allowing a resubmission of our manuscript, with an opportunity to address the reviewers’ comments.

We are uploading (a) our point-by-point response to the comments (below) (response to reviewers), (b) an updated manuscript with yellow highlighting indicating changes, and (c) a clean updated manuscript without highlights (PDF main document).

By following reviewers’ comments, we made substantial modifications in our paper to improve its clarity and readability. In our revised paper, we represent the improved manuscript.

We have done following modifications in short.

According to our 10 years of experience in Skin cancer, this is a novel study, which has been tested on computational devices as well.

(1) All reviewers’ comments related to abstract, introduction, methodology, contributions, results, comparisons, and discussions and conclusions.

(2) Literature review has been extended by adding new references and literature review with each study limitations.

(3) English writing has been improved with no grammatical mistakes in the paper.

(4) References has been written according to journal style.

We have made the following modifications as desired by the reviewers and editorial board, which are attached to this letter.

Best regards,

Corresponding Author,

Dr. Abdul Rauf Baig (On behalf of authors),

Dr. Qaisar Abbas

Professor.

Reviewer 2 Report (New Reviewer)

No further changes are required

Author Response

Original Manuscript ID diagnostics-2150020      

Original Article Title: Light-Dermo: A Lightweight Pretrained Convolution Neural Networks for Diagnosis of Pigmented Skin Lesions

To: Editor in Chief,

MDPI, Diagnostics

Re: Response to reviewers

Dear Editor,

Many thanks for insightful comments and suggestions of the referees. Thank you for allowing a resubmission of our manuscript, with an opportunity to address the reviewers’ comments.

We are uploading (a) our point-by-point response to the comments (below) (response to reviewers), (b) an updated manuscript with yellow highlighting indicating changes, and (c) a clean updated manuscript without highlights (PDF main document).

By following reviewers’ comments, we made substantial modifications in our paper to improve its clarity and readability. In our revised paper, we represent the improved manuscript.

We have done following modifications in short.

According to our 10 years of experience in Skin cancer, this is a novel study, which has been tested on computational devices as well.

(1) All reviewers’ comments related to abstract, introduction, methodology, contributions, results, comparisons, and discussions and conclusions.

(2) Literature review has been extended by adding new references and literature review with each study limitations.

(3) English writing has been improved with no grammatical mistakes in the paper.

(4) References has been written according to journal style.

We have made the following modifications as desired by the reviewers and editorial board, which are attached to this letter.

Best regards,

Corresponding Author,

Dr. Abdul Rauf Baig (On behalf of authors),

Dr. Qaisar Abbas

Professor.

This manuscript is a resubmission of an earlier submission. The following is a list of the peer review reports and author responses from that submission.

Round 1

Reviewer 1 Report

Show comparative analysis of results and discuss the same

Highlight the contributions and inferences drawn.

Include pathology of skin lesions with sample imagesof various classes

Author Response

Original Manuscript ID:  IDdiagnostics-2080443         

Original Article Title: LCNN-Dermo: A Lightweight Pretrained Convolution Neural Networks for Diagnosis of Pigmented Skin Lesions

To: Editor in Chief,

MDPI, Diagnostics

Re: Response to reviewers

Dear Editor,

Many thanks for insightful comments and suggestions of the referees. Thank you for allowing a resubmission of our manuscript, with an opportunity to address the reviewers’ comments.

We are uploading (a) our point-by-point response to the comments (below) (response to reviewers), (b) an updated manuscript with yellow highlighting indicating changes, and (c) a clean updated manuscript without highlights (PDF main document).

By following reviewers’ comments, we made substantial modifications in our paper to improve its clarity and readability. In our revised paper, we represent the improved manuscript.

We have made the following modifications as desired by the reviewers:

Best regards,

Corresponding Author,

Dr. Abdul Rauf Baig (On behalf of authors),

Professor.

Reviewer 2 Report

1.      The abstract and the introduction sections failed to support the problem statement. These sections of the article must identify the problem statement, such as the challenges to the existing techniques, and why the authors are interested in proposing this study as a solution.

2.      Authors have not reviewed/considered the latest published work because excellent work has already been published in this domain.

3.      The literature review is written haphazardly

4.      Material and Methods are presented in two sections, section 3 and section 4 with the same headings. Secondly, the text written in these sections should be part of related work if mandatory to mention, the text presented in these sections does not support the material and methods.

5.      Overall, there is a lack of flow throughout the paper. Hence it is difficult to grasp readers' attention.

6.      The organization of the paper is not up to the standard and the material is written haphazardly.

7.      The proposed methodology is very simple having no solid contribution.

8.      The study lacks a theoretical framework that is important for the reader to grasp the crust of the research.

9.      The experimental setup heading is given two times in Sections 5.1 and 5.2 which is inappropriate.

10.  Performance measures section again given 5.1. all performance measures are well known, therefore is no need to present them in detail

11.  The article is poorly formatted.

12.  Some of the references are incomplete and not in the same format

13.   There are lots of grammatical errors  and punctuation mistakes, therefore careful proofreading is required before publishing the article

In short, the overall impact of this article is not up to the mark with improper formatting, unclear and inappropriate flow of text and poor organization. There are other imperfections in this manuscript as well, but the above-mentioned deficiencies are sufficient to improve the draft.  

Author Response

(The authors gave the same response as above.)
